# FrontierCO: Real-World and Large-Scale Evaluation of Machine Learning Solvers for Combinatorial Optimization

**Shengyu Feng**[*]   **Weiwei Sun**[*]   **Shanda Li**   **Ameet Talwalker**   **Yiming Yang**
School of Computer Science, Carnegie Mellon University
{shengyuf, weiwes, shandal, atalwalk, yiming}@cs.cmu.edu

## Abstract

Machine learning (ML) has shown promise for tackling combinatorial optimization (CO), but much of the reported progress relies on small-scale, synthetic benchmarks that fail to capture real-world structure and scale. A core limitation is that ML methods are typically trained and evaluated on synthetic instance generators, leaving open how they perform on irregular, competition-grade, or industrial datasets. We present FrontierCO, a benchmark for evaluating ML-based CO solvers under real-world structure and extreme scale. FrontierCO spans eight CO problems, including routing, scheduling, facility location, and graph problems, with instances drawn from competitions and public repositories (e.g., DIMACS, TSPLib). Each task provides both easy sets (historically challenging but now solvable) and hard sets (open or computationally intensive), alongside standardized training/validation resources. Using FrontierCO, we evaluate **16 representative ML solvers**—graph neural approaches, hybrid neural–symbolic methods, and LLM-based agents—against state-of-the-art classical solvers. We find a persistent performance gap that widens under structurally challenging and large instance sizes (e.g., TSP up to 10M nodes; MIS up to 8M), while also identifying cases where ML methods outperform classical solvers. By centering evaluation on real-world structure and orders-of-magnitude larger instances, FrontierCO provides a rigorous basis for advancing ML for CO. Our benchmark is available at https://huggingface.co/datasets/CO-Bench/FrontierCO.

## 1 Introduction

Combinatorial optimization (CO) lies at the heart of computer science, operations research, and applied mathematics, with applications in routing, allocation, planning, and scheduling (Korte & Vygen, 2012). Most CO problems are intractable or NP-hard, and decades of research have relied on carefully engineered heuristics and exact solvers to make progress. Recently, machine learning (ML) has been proposed as a way to automate algorithm design, raising the exciting possibility that data-driven solvers could eventually rival or complement human-crafted methods.

Two main paradigms have emerged. *Neural solvers* use graph neural networks, reinforcement learning, or diffusion models to directly generate or guide solutions (Cappart et al., 2023; Bengio et al., 2020). *Symbolic solvers*, by contrast, leverage large language models (LLMs) to synthesize executable algorithms, often refining them through self-feedback or iterative search (Romera-Paredes et al., 2023; Liu et al., 2024; Ye et al., 2024; Novikov et al., 2025). Both paradigms have produced intriguing successes on benchmark datasets, sparking optimism about ML's role in CO.

Yet a central question remains unanswered: **can ML-based solvers match or surpass state-of-the-art (SOTA) human-designed algorithms on real-world CO problems?** Existing benchmarks do not allow us to answer this rigorously. They suffer from three limitations: (i) **scale:** most focus on toy instances orders of magnitude smaller than real applications (Kool et al., 2019; Luo et al., 2023); (ii) **realism:** synthetic datasets often fail to capture structural diversity; and (iii) **data realism and coverage**, i.e., most ML evaluations rely on synthetic generators, which limits insight

---

[*]Equal contributors.

into performance on irregular, non-Euclidean, or competition-grade instances that classical solvers routinely tackle. As a result, ML methods are often assessed at modest scales and on structurally simplified distributions.

To address these limitations, we present FRON-TIERCO, a benchmark that evaluates ML-based solvers under **real-world structure** and **extreme instance sizes** across eight CO problems from five categories (Figure 1). Unlike evaluations based solely on synthetic data, FRONTIERCO integrates instances from TSPLib, Reinelt (1991), DIMACS challenges (Johnson & McGeoch, 1993), CFLP testbeds (Avella et al., 2009), and other competition or repository sources, and complements them with standardized training/validation resources. For each problem we provide two test sets: easy (once challenging, now solvable by SOTA classical methods) and hard (open or computationally intensive). **We intentionally include structurally challenging cases** (e.g., PUC hypercubes (Rosseti et al., 2001); SAT-induced MIS (Xu et al., 2007)) and push scale by orders of magnitude to reflect real-world difficulty. Concretely, FRONTIERCO scales to TSP with 10M nodes and MIS with 8M nodes. Prior larger-scale ML evaluations (e.g., DIMES) scaled to TSP graphs with 10K nodes, while early neural TSP studies commonly used $\leq$ 100 nodes (Kool et al., 2019).

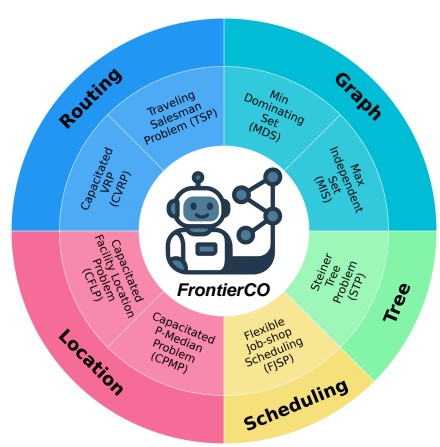

Figure 1: Overview of FRONTIERCO.

Using this benchmark, we conduct a systematic, cross-paradigm evaluation of ML-based CO solvers. Our study covers 16 representative approaches, including end-to-end neural solvers, neural-enhanced heuristics (Bengio et al., 2020; Cappart et al., 2023), and LLM-based agentic methods (Sun et al., 2025), and compares them directly against the best human-designed solvers. This unified evaluation reveals several key insights: (i) ML methods still lag significantly behind SOTA human solvers, especially on hard instances; (ii) neural solvers demonstrate the potential to enhance simple human heuristics, but in general struggle with scalability, non-local structure, and distribution shift; (iii) LLM-based solvers sometimes may outperform the SOTA classical solvers but display high variance due to their incapability in understanding the effectiveness of different algorithms they are trained on.

Our contributions are threefold.

1. **Benchmark under real-world structure and extreme scale.** A unified evaluation suite across eight problems that pairs competition/real-world instances with hard, structurally irregular cases and orders-of-magnitude larger sizes than prior ML evaluations (e.g., TSP: 10M vs. 10k; MIS: 8M vs. 11k (Qiu et al., 2022)).

2. **Unified evaluation.** We conduct a rigorous comparison of 16 ML-based solvers against state-of-the-art classical baselines, under standardized protocols.

3. **Empirical insights.** We identify fundamental limitations of current ML approaches, while also highlighting the potential and future research directions for ML-based solvers.

## 2 FRONTIERCO: THE PROPOSED BENCHMARK

### 2.1 FORMAL OBJECTIVE AND EVALUATION METRICS

We follow Papadimitriou & Steiglitz (1982) in denoting a combinatorial optimization (CO) problem instance as $s$, a solution as $x \in \mathcal{X}_s$, and defining the objective as

$$\min_{x \in \mathcal{X}_s} c_s(x) = \text{cost}(x; s) + \text{valid}(x; s), \tag{1}$$

where $\text{cost}(x; s)$ is a problem-specific objective (e.g., the tour length in routing problems), and $\text{valid}(x; s)$ penalizes constraint violations—taking value $\infty$ if $x$ is infeasible, and 0 otherwise. Note

that any maximization problem can be turned into a minimization one by negating the objective sign, and we treat all problems in its minimization version for unified evaluation in this work.

To accommodate the varying scales of different problem instances, we define the *primal gap* as:

$$\text{pg}(x; s) = \begin{cases} 1, & \text{if } x \text{ is infeasible or } \text{cost}(x; s) \cdot c^* < 0, \\ \dfrac{|\text{cost}(x; s) - c^*|}{\max\{|\text{cost}(x; s)|, |c^*|\}}, & \text{otherwise,} \end{cases} \tag{2}$$

where $c^*$ is the (precomputed) optimal or best-known cost for instance $s$. This metric has been popularly used in classical solvers (Berthold, 2006; 2013; Achterberg et al., 2012), the DIMACS challenge (DIM, 2013-2014), and recent neural solvers (Nair et al., 2020; Chmiela et al., 2021; Huang et al., 2023). By definition, our primal gap is strictly bounded within the range $[0, 1]$ (i.e., 0% to 100%)[1], where 0 is optimal and 1 is the worst possible score. Any feasible solution will result in a gap strictly less than 1. We set the primal gap for infeasible solutions to 1 to flag them as failures, aligning with the intuition that infeasible solutions are never better than feasible ones.

## 2.2 DOMAIN COVERAGE

This study focuses on eight types of CO problems that have gained increasing attention in recent machine learning research. These problems are:

- **MIS (Maximum Independent Set)**: Find the largest subset of non-adjacent vertices in a graph, whose minimization version (corresponding to Equation 1) is to minimize the negative set size.
- **MDS (Minimum Dominating Set)**: Find the smallest subset of vertices such that every vertex in the graph is either in the subset or adjacent to a vertex in the subset.
- **TSP (Traveling Salesman Problem)**: Find the shortest possible tour that visits each city exactly once and returns to the starting point. We focus on the 2D Euclidean space in this work.
- **CVRP (Capacitated Vehicle Routing Problem)**: Determine the optimal set of delivery routes for a fleet of vehicles with limited capacity to serve a set of customers.
- **CFLP (Capacitated Facility Location Problem)**: Choose facility locations and assign clients to them to minimize the total cost, subject to facility capacity constraints.
- **CPMP (Capacitated $p$-Median Problem)**: Select $p$ facility locations and assign clients to them to minimize the total distance, while ensuring that no facility exceeds its capacity.
- **FJSP (Flexible Job-Shop Scheduling Problem)**: Schedule a set of jobs on machines where each operation can be processed by multiple machines, aiming to minimize the makespan while respecting job precedence and machine constraints.
- **STP (Steiner Tree Problem)**: Find a minimum-cost tree that spans a given subset of terminals in a graph, possibly including additional intermediate nodes.

The dataset statistics are summarized in Table 1, with additional details provided in the Appendix B. Note that only test data are collected from the listed sources; training and validation data generated from the same synthetic generator to ensure they are from the same distribution (but may at different scales and set size dependent on the model efficiency/scalability), in order to ensure the fair comparison among neural and LLM solvers (see Section 2.5).

Graph-based problems (MIS and MDS) and routing problems (TSP and CVRP) have been widely used to evaluate end-to-end neural solvers (Qiu et al., 2022; Zhang et al., 2023; Sun & Yang, 2023; Sanokowski et al., 2025), as these tasks often admit relatively straightforward decoding strategies to transform probabilistic model output into feasible solutions. In contrast, facility location and scheduling problems (such as CFLP, CPMP, and FJSP) involve more complex and interdependent constraints, making them better suited to hybrid approaches that combine neural networks with traditional solvers (Gasse et al., 2019; Scavuzzo et al., 2022; Feng & Yang, 2025b). Tree-based problems have received comparatively less attention in neural CO, yet we include a representative case (e.g., STP) due to their fundamental importance in the broader CO landscape. All of the above problems can also be directly handled by symbolic solvers, enabling comprehensive and comparable evaluations across solver paradigms (Romera-Paredes et al., 2023; Liu et al., 2024; Ye et al., 2024).

---

[1]Note that this gap definition is different from the one widely used in routing literature (Kool et al., 2019; Sun & Yang, 2023; Luo et al., 2023), which can be larger than 100%.

Table 1: Summary of collected problem instances.

| Problem | Test Set Sources | Attributes | Easy Set | Hard Set |
|---|---|---|---|---|
| MIS | 2nd DIMACS Challenge
BHOSLib | Instances
Nodes | 36
1,404–7,995,464 | 16
1,150–4,000 |
| MDS | PACE Challenge 2025 | Instances
Nodes | 20
2,671–675,952 | 20
1,053,686–4,298,062 |
| TSP | TSPLib
8th DIMACS Challenge | Instances
Cities | 29
1,002–18,512 | 19
10,000–10,000,000 |
| CVRP | Golden et al. (1998)
Arnold et al. (2019) | Instances
Cities | 20
200–483 | 10
3,000–30,000 |
| CFLP | Avella & Boccia (2009)
Avella et al. (2009) | Instances
Facilities
Customers | 20
1,000
1,000 | 30
2,000
2,000 |
| CPMP | Lorena & Senne (2004; 2000)
Stefanello et al. (2015)
Gnägi & Baumann (2021) | Instances
Facilities
Medians | 31
100–4,461
10–1,000 | 12
10,510–498,378
100–2,000 |
| FJSP | Behnke & Geiger (2012)
Naderi & Roshanaei (2021) | Instances
Jobs
Machines | 60
10–100
10–20 | 20
10–100
20–60 |
| STP | Leitner et al. (2014)
Rosseti et al. (2001) | Instances
Nodes | 23
7,565–71,184 | 50
64–4,096 |

## 2.3 PROBLEM INSTANCES

For each CO problem type, we collect a diverse pool of problem instances from problem-specific and comprehensive CO libraries (Reinelt, 1991; Xu et al., 2007), major CO competitions (Johnson & McGeoch, 1993; PACE, 2025), and evaluation sets reported in recent research papers.

Due to rapid progress in CO, many instances from earlier archives can now be effectively solved by SOTA problem-specific solvers, often achieving an optimality gap below 1% within a 1-hour time budget. We select a representative subset of such instances as our *easy set*, which serves to validate the baseline effectiveness of ML-based solvers.

With a high-level goal to advance the CO solvers on open challenges, we also construct a *hard set* comprising open benchmark instances widely used to assess cutting-edge human-designed algorithms. Many of these instances lack known optimal solutions and remain beyond the reach of existing heuristics. As a result, they are less susceptible to *heuristic hacking*, where neural solvers or LLM-based agents rely on handcrafted decoding strategies or memorize prior solutions, rather than learning to solve the problem from first principles. Importantly, our hard set is not defined merely by instance size. Instead, we emphasize structurally complex cases, such as hypercube graphs in STP (Rosseti et al., 2001) or SAT-induced MIS (Xu et al., 2007), which require models to understand and reason about intricate problem structures.

## 2.4 SOTA SOLVERS AND BEST KNOWN SOLUTIONS (BKS)

We identify the SOTA solver for each CO problem type based on published research papers and competition leaderboards. The selected solvers include: KaMIS (Lamm et al., 2017) for MIS, LKH-3 (Helsgaun, 2017) for TSP, HGS (Vidal et al., 2012) for CVRP, GB21-MH (Gnägi & Baumann, 2021), a hybrid metaheuristic, for CPMP, and SCIP-Jack (Rehfeldt et al., 2021) for STP. For problems where no dominant problem-specific solver is available (e.g., MDS, CFLP, FJSP), we rely on general-purpose commercial solvers, such as Gurobi (Gurobi Optimization, LLC, 2024) for MDS and CFLP (Mixed Integer Programming), and CPLEX (Cplex, 2009) for FJSP (Constraint Programming). Among them, Gurobi, CPLEX and SCIP-Jack are exact solvers; the rest are heuristic-based.

Prior evaluations of ML-based CO solvers often relied on self-generated synthetic test instances, leading to difficulties in fair comparison across papers. These instances are sensitive to imple-

mentation details such as random seeds and Python versions, introducing undesirable variability and inconsistency. To address this, we provide standardized BKS for all test-set instances in our benchmark. These BKS are collected from published literature and competition leaderboards, and are further validated using the corresponding SOTA solvers executed on our servers. For instances lacking known BKS, such as the MDS instances from the PACE Challenge 2025 (PACE, 2025), or for benchmarks with outdated references, such as those in the CFLP literature, we run the designated SOTA solver for up to two hours to obtain high-quality reference solutions.

## 2.5 STANDARDIZED TRAINING/VALIDATION DATA

Similar to BKS, inconsistencies in self-generated training and validation data can also contribute to difficulties in cross-paper comparisons. To address this, FRONTIERCO provides standardized training sets for neural solvers and development sets for LLM agents, generated using a variety of problem-specific instance generators (details in Appendix B).

We also release a complete toolkit that includes a data loader, an evaluation function, and an abstract solving template tailored for LLM-based agents. The data loader and evaluation function are hidden from the agents to prevent data leakage. The solving template provides a natural language problem description along with Python starter code specifying the expected input and output formats. An example prompt is provided in Appendix C.3.

## 3 EVALUATION DESIGN

### 3.1 IMPLEMENTATION SETTINGS

In light of the difficulty and scale of our problem instances, we allow a maximum solving time of one hour per problem instance, as most solvers, including both classical and ML-based solvers, may require such a time to obtain a single feasible solution (see efficiency analysis in Appendix E).

For fair comparison, each solver is executed on a single CPU core of a dual AMD EPYC 7313 16-Core processor, and neural solvers are run on a single NVIDIA RTX A6000 GPU. Since the solving time is influenced by factors such as compute hardware (CPU vs. GPU), solver type (exact vs. heuristic), and implementation language (C++ vs. Python), **we use the primal gap (Equation 2) as the primary evaluation metric, and solving time is reported for reference only**. For any infeasible solution, we assign a primal gap of 1 and a solving time of 3600 seconds. The arithmetic mean of the primal gaps and geometric mean of solving time are reported across our experiments.

### 3.2 REPRESENTATIVE NEURAL SOLVERS FOR COMPARATIVE EVALUATION

In addition to the SOTA human-designed solvers described in Section 2.4, we include a curated set of machine learning-based CO solvers from recent literature. The neural solvers are tailored to specific problem categories they are developed for:

- **DiffUCO** (Sanokowski et al., 2024): An unsupervised diffusion-based neural solver for MIS and MDS that learns from the Lagrangian relaxation objective.
- **SDDS** (Sanokowski et al., 2025): A more scalable version of DiffUCO for MIS and MDS, with efficient training process.
- **RLNN** (Feng & Yang, 2025a): A neural sampling framework that enhances exploration in CO by enforcing expected distances between sampled and current solutions.
- **LEHD** (Luo et al., 2023): A hybrid encoder-decoder model for TSP and CVRP, with strong generalization to real-world instances.
- **DIFUSCO** (Sun & Yang, 2023): A diffusion-based approach for TSP that achieves strong scalability, solving instances with up to 10,000 cities.
- **SIL** (Luo et al., 2023): A linear-complexity transformer solver that achieves extreme scalability, handling routing instances with up to 100,000 cities.
- **DeepACO** (Ye et al., 2023): A neural solver that adapts Ant Colony Optimization (ACO) principles to learn metaheuristic strategies.

- **tMDP** (Scavuzzo et al., 2022): A reinforcement learning framework that models the branching process in Mixed Integer Program (MIP) solver as a tree-structured Markov Decision Process.
- **SORREL** (Feng & Yang, 2025b): A reinforcement learning method that leverages suboptimal demonstrations and self-imitation learning to train branching policies in MIP solvers.
- **GCNN** (Gasse et al., 2019): A graph convolutional network (GNN)-guided solver for MIPs, which learns to guide branching decisions within a branch-and-bound framework.
- **IL-LNS** (Sonnerat et al., 2021): A neural large neighborhood search method for Integer Linear Programs (ILPs) that is trained to predict the locally optimal neighborhood choice.
- **CL-LNS** (Huang et al., 2023): A contrastive learning-based large neighborhood search approach for ILPs which advances the imitation learning strategy in IL-LNS.
- **MPGN** (Lei et al., 2022): A reinforcement learning-based approach for FJSP that employs multi-pointer graph networks to capture complex dependencies and generate efficient schedules.
- **L-RHO** (Li et al., 2025a): A learning-guided rolling horizon optimization method that integrates machine learning predictions into the rolling horizon framework.

Since STP is not well studied by existing neural methods, we consider both reinforcement learning (**RL**) and supervised learning (**SL**) baselines, predicting the Steiner points. The Takahashi–Matsuyama algorithm (Takahashi & Matsuyama, 1980) is then applied for decoding.

### 3.3 REPRESENTATIVE LLM-BASED AGENTS FOR COMPARATIVE EVALUATION

Our LLM-based solvers are selected based on the CO-Bench evaluation protocol (Sun et al., 2025), including both general-purpose prompting approaches and CO-specific iterative strategies:

- **FunSearch** (Romera-Paredes et al., 2023): An evolutionary search framework that iteratively explores the solution space and refines candidates through backtracking and pruning.
- **Self-Refine** (Madaan et al., 2023; Shinn et al., 2023): A feedback-driven refinement method in which the LLM improves its own output via iterative self-refinement.
- **ReEvo** (Ye et al., 2024): A self-evolving agent that leverages past trajectories—both successful and failed—to refine its future decisions through reflective reasoning.

All LLM-based solvers are evaluated across the full set of eight CO problem types in our benchmark.

## 4 RESULTS

We summarize the comparative results in Figure 2 and Table 2. See detailed results in Appendix D. **Note that the primal gap is computed relative to the best known solution (BKS), so its absolute value does not directly reflect the inherent difficulty of the instance**—especially in cases where no known optimum exists.

We draw several key observations from our results. **First, there is a substantial performance gap between human-designed state-of-the-art (SOTA) solvers and ML-based solvers** across all problem types and difficulty levels. Strikingly, this gap is more pronounced in our benchmark than in previously published results. For instance, LEHD reports only a 0.72% gap on a standard TSP benchmark (Kool et al., 2019), whereas on our new benchmark the gap widens to 10% on easy TSP instances and an alarming 77% on hard instances. A major factor behind this discrepancy lies in the training and evaluation protocols. Prior studies typically trained neural solvers on synthetic graphs of a fixed size (e.g., 1000 nodes) and evaluated them on test instances of the same size, ensuring aligned conditions. In contrast, our datasets incorporate substantial variability in both graph size and structure across training and test sets. This setup better reflects real-world deployment scenarios but also introduces significant distribution shifts, under which LEHD and many other ML-based methods experience severe performance degradation in FRONTIERCO.

**Second, neural solvers face serious scalability challenges.** Although they used to be treated as efficient heuristics on large-scale, difficult instances, we find that in practice this is often not the case. Neural networks typically address the non-convexity of CO problems through over-parameterization

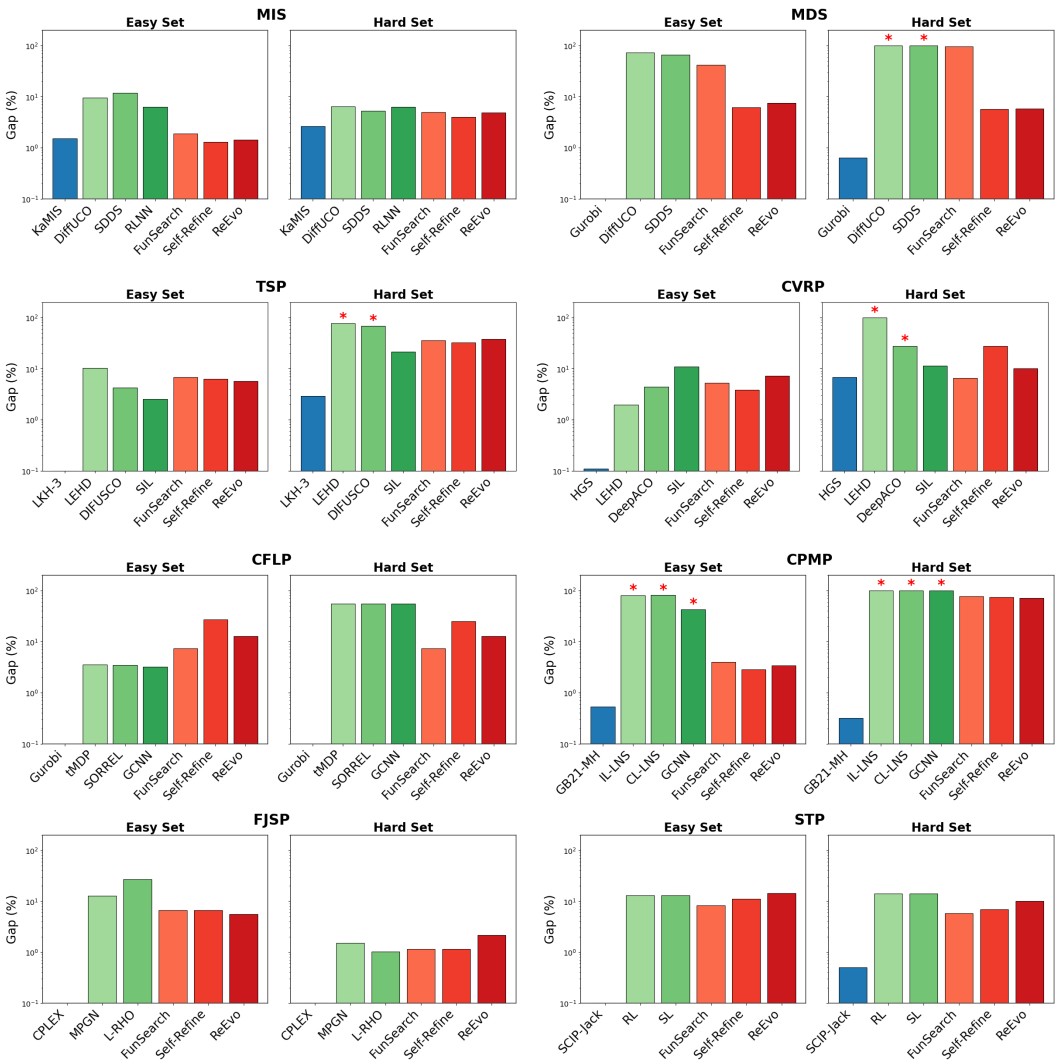

Figure 2: Primal gap (%) across eight CO problems on easy and hard sets (lower is better). Classical (blue), neural (green), and LLM-based agents (red). Bars marked with * indicate at least one infeasible run on that test set; in such cases we assign gap 1 and time 3600 seconds (see Section 3.1).

(Allen-Zhu et al., 2019), which inflates single-value variables into high-dimensional representations and leads to frequent out-of-memory failures (observed in 4 of 8 problems; see Appendix D). Inference efficiency is an additional bottleneck. For example, the auto-regressive solver LEHD (Luo et al., 2023) requires running a transformer model (Vaswani et al., 2017) for 10M steps to produce a single solution on our largest TSP instance, failing to return any solution within the 1-hour time limit. Similar inefficiencies exist even on easier instances or under shorter time budgets (Appendix E). Addressing these issues through integration of reduction techniques (Andersen & Andersen, 1995) and the design of more compact neural architectures is thus an important direction for future research.

**Third, LLM-based agents show the potential to outperform prior human-designed SOTA solvers.** For example, Self-Refine surpasses KaMIS on the easy MIS set, and FunSearch outperforms HGS on the hard CVRP set. A closer inspection of these methods reveals their algorithmic sophistication: Self-Refine applies kernelization to simplify MIS instances, solves small kernels exactly using a Tomita-style max-clique algorithm, and employs ARW-style heuristics with solution pools, crossover, and path-relinking for larger instances. Similarly, FunSearch builds an Iterated Local Search framework for CVRP, enhanced with regret insertion and Variable Neighborhood Descent. These results highlight

the promise of LLM-based approaches in automatically developing competitive, and in some cases superior, solvers for CO.

**Fourth, despite their promise, LLM-based agents exhibit substantial performance variability.** For example, while they perform comparably to the SOTA solver HGS on the hard CVRP set, they fall dramatically short on TSP—even though both are routing problems. We hypothesize that this stems from the nature of LLM training: while models are exposed to diverse human-designed heuristics and can combine them in novel ways, they generally lack the ability to reliably assess the effectiveness of the generated algorithms. As a result, each sampling run may randomly yield a different, not necessarily effective, strategy. This absence of internal reasoning abilities largely restricts the applicability of LLM agents to hard-to-verify tasks and raises safety concerns when they generate resource-intensive algorithms for large instances (e.g., frequent out-of-memory issues on CPMP during evolving). Current agentic frameworks tend to focus on problems that are challenging yet easy to verify, strongly relying on external feedback. In contrast, FRONTIERCO provides a hard-to-verify benchmark (but still verifiable for evaluation purposes) that highlights the reasoning capabilities of the LLM themselves.

Table 2: The average primal gap achieved by LLM agentic solvers over all eight CO problems.

| Method | Avg. Gap ↓ (All) | Avg. Gap ↓ (Easy) | Avg. Gap ↓ (Hard) |
|---|---|---|---|
| FunSearch | 20.35% | 10.05% | 30.65% |
| Self-Refine | 15.11% | 8.18% | 22.03% |
| ReEvo | **13.25%** | **7.25%** | **19.25%** |

Table 3: Ablation study on the effectiveness of the neural module.

| TSP-Easy | | CFLP-Easy | |
|---|---|---|---|
| Method | Gap ↓ | Method | Gap ↓ |
| LKH-3 | **0.03**% | Gurobi | **0.00**% |
| 2-OPT | 20.09% | SCIP | 6.50% |
| DIFUSCO | 4.19% | GCNN | 3.22% |

## 5 DISCUSSIONS

### 5.1 DOES THE NEURAL MODULE HELP?

Considering the performance gap between neural solvers and SOTA solvers, a natural question arises: does the neural module actually contribute to improved performance? To explore this, we conduct an ablation study by removing the neural component from the underlying algorithm of each neural solver. We evaluate two representative pairs: DIFUSCO (Sun & Yang, 2023) vs. 2-OPT, and GCNN (Gasse et al., 2019) vs. SCIP (Achterberg, 2009). The results are summarized in Table 3.

The results show that both DIFUSCO and GCNN significantly improve upon their respective heuristic baselines, indicating a meaningful contribution from the neural module. However, such improvement is still far from being comparable to the SOTA classical solvers. Overall, our findings suggest that **neural components can enhance human-designed heuristics, but such improvement is typically realized when built on relatively weak base algorithms**. Whether similar gains can be achieved when enhancing already strong heuristics remains unclear.

### 5.2 DO NEURAL SOLVERS CAPTURE GLOBAL STRUCTURE?

Most neural solvers are based on graph neural networks (GNNs)[2], which rely on local message passing. While they have demonstrated strong performance on routing problems such as TSP and CVRP—which involve complex global constraints—the majority of existing evaluations are limited to 2D Euclidean instances. Compared to general graph problems, Euclidean instances—such as those in metric TSP—often exhibit favorable local structures (e.g., triangle inequality), which can be explicitly exploited by certain algorithms to achieve improved performance (Karlin et al., 2021). In contrast, general graph problems such as MIS lack such spatial regularities, and neural solvers often perform poorly on them (Angelini & Ricci-Tersenghi, 2022; Böther et al., 2022).

To explicitly evaluate the ability of neural solvers in capturing global structure, we leverage the rich source of STP instances, which includes both Euclidean and non-Euclidean graphs (see Appendix B.8

---

[2]By GNN, we refer to general message passing frameworks including attention-based neural architectures.

for details). We train two separate GNNs to predict Steiner nodes, using ground truth labels generated by SCIP-Jack (Rehfeldt et al., 2021). One model is trained on Euclidean instances, and the other on non-Euclidean instances. The training dynamics are shown in Figure 3.

The results reveal a clear contrast: **while the GNN quickly achieves a high F1 score in predicting Steiner points on Euclidean graphs, it fails to make any progress on non-Euclidean ones.** This suggests that existing GNNs implicitly rely on locality and cannot really capture the global structure. These findings underscore a fundamental limitation in the expressive power of current neural solvers. In Appendix F, we also extend the discussion to the non-Euclidean (non-metric) TSP instances.

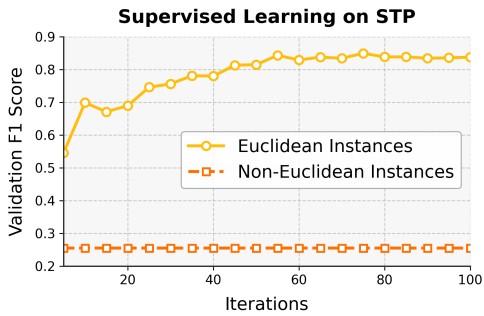

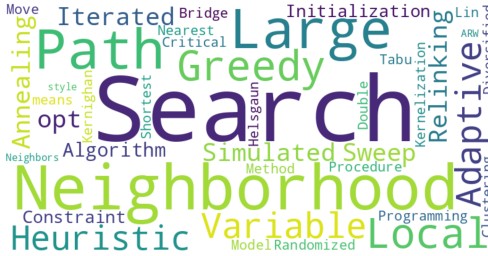

Figure 3: Training dynamics of neural solvers on Euclidean and non-Euclidean STP instances.

Figure 4: Word cloud of the algorithms generated by LLM-based solvers.

### 5.3 WHAT KINDS OF ALGORITHMS DO LLM-BASED SOLVERS DISCOVER?

To better understand the algorithmic strategies developed by LLM-based solvers, we visualize the key words corresponding to their generated algorithms using the word cloud in Figure 4, where the size of each word reflects its frequency of appearance across algorithms.

A clear pattern emerges: classical metaheuristics—particularly simulated annealing (SA) and large neighborhood search (LNS)—consistently appear across a diverse set of problems and often form the foundation of LLM-generated algorithms. **This highlights a shared reliance on well-established CO algorithms that effectively balance exploration and exploitation.** While current LLMs still fall short of demonstrating novel algorithmic reasoning (algorithms that cannot be mapped to existing ones) in CO, their strategies tend to replicate known metaheuristics and problem-specific techniques from the literature. Interestingly, we observe that their performance does not critically depend on integrating existing solvers, suggesting that LLMs can autonomously construct plausible and often effective algorithms. This adaptability is particularly promising for rapidly tackling new problem variants or classical problems with additional constraints, indicating strong potential for LLMs in zero-shot or few-shot algorithm design scenarios.

## 6 RELATED WORK

Current machine-learning approaches to CO fall into two broad categories: neural and symbolic solvers. Neural solvers primarily train a graph neural network (GNN) model with standard machine learning objectives (Bengio et al., 2020; Cappart et al., 2023). The trained GNN is then used either to predict complete solutions (Luo et al., 2023; Sun & Yang, 2023; Sanokowski et al., 2024; 2025) or to guide classical heuristics such as branch-and-bound (Gasse et al., 2019; Scavuzzo et al., 2022; Feng & Yang, 2025b) and large neighborhood search (Sonnerat et al., 2021; Huang et al., 2023; Feng et al., 2025). Symbolic solvers instead attempt to generate executable programs that solve the problem, exploring the space of algorithmic primitives with reinforcement learning (Kuang et al., 2024a;b) or leveraging LLM agents for code generation (Romera-Paredes et al., 2023; Ye et al., 2024; Liu et al., 2024; Novikov et al., 2025).

Despite these advances, empirical studies have mostly focused on synthetic benchmarks (Kool et al., 2019; Zhang et al., 2023; Berto et al., 2025; Bonnet et al., 2024; Ma et al., 2025), falling short in

scalability and diversity, or restricted to a single type of CO problems (Thyssens et al., 2023; Li et al., 2025b). Besides, the lack of training instances in existing LLM agentic benchmarks (Fan et al., 2024; Tang et al., 2025; Sun et al., 2025) also hinders the further development. To bridge these gaps, we introduce a comprehensive benchmark with both realistic evaluation instances and diverse training data sources.

## 7 CONCLUSION

We present FRONTIERCO, a new benchmark designed to rigorously evaluate ML-based CO solvers under realistic, large-scale, and diverse problem settings. Through a unified empirical study, we reveal that while current ML methods—including both neural and LLM-based solvers—show potential, they still lag behind state-of-the-art human-designed algorithms in terms of efficiency, generalization, and scalability. However, our findings also uncover promising avenues: neural solvers excel on structured problems, and LLM agents demonstrate novel strategy discovery on hard instances. We hope FRONTIERCO will serve as a foundation for advancing the design and evaluation of next-generation ML-based CO solvers.

**Boarder Impact**  FRONTIERCO offers a standardized, challenging benchmark to advance ML for combinatorial optimization. It enables rigorous, reproducible evaluation, encourages scalable and generalizable solver development, and highlights current limitations to guide more robust and impactful AI solutions in real-world decision-making.

## REPRODUCIBILITY STATEMENT

Details of data collection are provided in Appendix B. The implementations of neural solvers are taken from the official public repositories of each method, as referenced in Section 3.2. All remaining code, including that for classical solvers, BKS computation, and LLM agent solvers, is available at https://github.com/sunnweiwei/FrontierCO.

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

## A  THE USE OF LARGE LANGUAGE MODELS

Large Language Models (LLMs) were used exclusively for supportive purposes, such as adapting baseline implementations, processing data, generating plots, and refining the manuscript text. Importantly, LLMs were not involved in data collection/synthesis, experimental design and result analysis, and therefore did not influence the scientific contributions of this work.

## B  DATA COLLECTION DETAILS

This section outlines the data collection process for all problems, covering both test and training/validation instances. Since the training instance generation for neural solvers varies significantly across methods, we omit low-level details such as the number of instances and parameter settings. **Instead, we focus on describing the generation of the validation set (test cases used to provide feedback for iterative agent refinement) used for LLM-based solvers.**

### B.1  MAXIMUM INDEPENDENT SET

To construct suitable test instances, we conduct a comprehensive re-evaluation of the datasets collected by Böther et al. (2022). We find that some large real-world graphs (Leskovec & Krevl, 2014), such as ai-caida (Leskovec et al., 2005) with up to 26,475 nodes, are not particularly challenging for SOTA classical solvers like KaMIS (Lamm et al., 2017), which can solve them within seconds. Therefore, we select two moderately sized but more challenging datasets.

The easy test set comprises complementary graphs of the maximum clique instances from the 2nd DIMACS Challenge (Johnson & Trick, 1996), while the hard test set consists of the largest 16 instances (each with over 1,000 nodes) from the BHOSLib benchmark (Xu et al., 2007), derived from SAT reductions. Since the original links have expired, we obtain these instances and their BKS from a curated mirror[3]. For those interested in additional sources of high-quality MIS instances, we also highlight vertex cover instances from the 2019 PACE Challenge[4], reductions from coding theory[5], and recent constructions derived from learning-with-errors (LWE) (Kawano, 2023), which provide a promising strategy for generating challenging MIS instances.

Training instances are generated using the RB model (Xu & Li, 2000), widely adopted in recent neural MIS solvers (Zhang et al., 2023; Sanokowski et al., 2024; 2025). We synthesize 20 instances with 800–1,200 nodes for our LLM validation set.

### B.2  MINIMUM DOMINATING SET

Despite the popularity of MDS in evaluating neural solvers (Zhang et al., 2023; Sanokowski et al., 2024; 2025), we find a lack of high-quality publicly available benchmarks. We therefore rely on the PACE Challenge 2025[6], using the exact track instances as our easy set and the heuristic track instances as the hard set. From each, we selected the 20 instances with the highest primal-dual gaps after a one-hour run with Gurobi. Reference BKS are obtained by extending the solving time to two hours.

---

[3]`https://iridia.ulb.ac.be/~fmascia/maximum_clique/`
[4]`https://pacechallenge.org/2019/`
[5]`https://oeis.org/A265032/a265032.html`
[6]`https://pacechallenge.org/2025/`

Training instances are Barabási–Albert graphs (Barabási & Albert, 1999) with 800–1,200 nodes, consistent with previous literature (Zhang et al., 2023; Sanokowski et al., 2024; 2025). We generate 20 such instances for the LLM validation set.

### B.3   TRAVELING SALESMAN PROBLEM

We source TSP instances from the 8th DIMACS Challenge[7] and TSPLib[8]. The easy test set includes symmetric 2D Euclidean TSP instances (distance type `EUC_2D`, rounding applied) from TSPLib with over 1,000 cities, all with known optimal solutions. This aligns with settings used in prior neural TSP solvers (Karlin et al., 2021).

The hard test set consists of synthetic instances from the DIMACS Challenge with at least 10,000 cities (Fu et al., 2023). We obtain BKS from the LKH website[9].

Training instances follow the standard practice of uniformly sampling points in a unit square (Kool et al., 2019). For simplicity, we reuse DIMACS instances with 1,000 nodes as our LLM validation set, since they are drawn from the same distribution, except scaling the coordinates by a constant.

### B.4   CAPACITATED VEHICLE ROUTING PROBLEM

We collect CVRP instances from the 12th DIMACS Challenge[10] and CVRPLib[11], which have significant overlap. From these, we select the Golden (Golden et al., 1998) and Belgium (collected by Arnold et al. (Arnold et al., 2019)) instances as our easy and hard sets, respectively. We discard the route length constraints in the first eight Golden instances in our experiments. All BKS are retrieved from the CVRPLib website.

Training data generation follows the method used in DeepACO (Ye et al., 2023). Each instance includes up to 500 cities, with demands in [1, 9] and capacity fixed at 50. We generate 15 total validation instances for LLMs, with 5 each for 20, 100, and 500 cities.

### B.5   CAPACITATED FACILITY LOCATION PROBLEM

Following the benchmark setup in previous works (Guastaroba & Speranza, 2012; Caserta & Voß, 2020), we select instances from Test Bed 1 (Avella & Boccia, 2009) and Test Bed B (Avella et al., 2009) as our easy and hard test sets, respectively. The easy set includes the 20 largest instances from Test Bed 1, each with 1,000 facilities and 1,000 customers. The hard set consists of the 30 largest instances from Test Bed B, each with 2,000 facilities and 2,000 customers. All instances are downloaded from the OR-Brescia website[12].

Notably, our easy instances are already significantly larger than the most challenging instances typically used in neural solver evaluations (Gasse et al., 2019; Scavuzzo et al., 2022; Feng & Yang, 2025b), which contain at most 100 facilities and 400 customers. All easy instances can be solved exactly by Gurobi. For the hard instances, as all available BKS identified in the literature (Caserta & Voß, 2020) are inferior to those obtained by Gurobi, we rerun Gurobi for two hours to obtain improved reference solutions.

Overall, we find that Gurobi already demonstrates strong performance on standard CFLP variants, in which each customer may be served by multiple facilities. Consequently, the single-source CFLP variant—where each customer must be assigned to exactly one facility—has become a more compelling and actively studied problem in recent CO literature (Gadegaard et al., 2017; Caserta & Voß, 2020; Almeida et al., 2023). Several corresponding benchmarks are also available on the OR-Brescia website.

For training data, we adopt the synthetic generation method from Cornuejols et al. (Cornuejols et al., 1991), producing 20 instances with 100 facilities and 100 customers for LLM validation. This

---

[7]http://archive.dimacs.rutgers.edu/Challenges/TSP/
[8]http://comopt.ifi.uni-heidelberg.de/software/TSPLIB95/
[9]http://webhotel4.ruc.dk/~keld/research/LKH/DIMACS_results.html
[10]http://dimacs.rutgers.edu/programs/challenge/vrp/cvrp/
[11]http://vrp.galgos.inf.puc-rio.br/index.php/en/
[12]https://or-brescia.unibs.it/home

generation method is widely used in existing neural branching works (Gasse et al., 2019; Scavuzzo et al., 2022; Feng & Yang, 2025b), and forms part of the construction for Test Bed 1 (Avella & Boccia, 2009).

## B.6 CAPACITATED $p$-MEDIAN PROBLEM

We follow the evaluation setup in recent works on CRMP (Stefanello et al., 2015; Gnägi & Baumann, 2021). Instances with fewer than 10,000 facilities are assigned to the easy set; larger ones go to the hard set. Easy instances include 6 real-world São José dos Campos instances (Lorena & Senne, 2004) and 25 adapted TSPLib instances (Lorena & Senne, 2000; Stefanello et al., 2015). These are sourced from INPE[13] and SomAla[14] websites. Hard instances are large-scale problems introduced by Gnägi and Baumann (Gnägi & Baumann, 2021), downloaded from their GitHub[15]. BKS are derived by combining the best GB21-MH results and values reported in (Stefanello et al., 2015; Steglich, 2019; Gnägi & Baumann, 2021).

In total, we collect 31 easy and 12 hard instances, all using Euclidean distances. Additional alternatives include spherical-distance instances (Diaz & Fernandez, 2006; Statistisches Bundesamt, 2017) and high-dimensional instances (Gnägi & Baumann, 2021).

We synthesize training data with Osman's method (Osman, 1994). The validation set for LLMs are generated by fixing the number of facilities at 500 and varying medians $p$ in $\{5, 10, 20, 50\}$. Each setting includes 5 instances.

## B.7 FLEXIBLE JOB-SHOP SCHEDULING PROBLEM

We collect FJSP instances from two recent benchmark sets commonly used in the evaluation of classical FJSP solvers. The easy test set consists of instances introduced by Behnke and Geiger (Behnke & Geiger, 2012), available via a GitHub mirror[16]. The hard test set includes 24 of the largest instances (with 100 jobs) from a benchmark proposed by Naderi and Roshanaei (Naderi & Roshanaei, 2021), which we obtain from the official repository[17]. These two datasets are selected based on recent comparative studies in the literature (Bahman Naderi, 2023; Dauzère-Pérès et al., 2024).

Based on our literature review, the strongest results have been reported by the CP-based Benders decomposition method (Naderi & Roshanaei, 2021); however, the source code is not publicly available. As a result, we adopt a constraint programming approach using CPLEX, which has demonstrated consistently strong performance relative to other commercial solvers and heuristic methods (Bahman Naderi, 2023).

Training data is generated following the same protocol used in Li et al. (Li et al., 2025a). Specifically, we synthesize 20 instances, each with 20 machines and 10 jobs, to form the LLM validation set.

## B.8 STEINER TREE PROBLEM

We collect STP instances from SteinLib[18] and the 11th DIMACS Challenge[19]. The easy set includes Vienna-GEO instances (Leitner et al., 2014), which—despite having tens of thousands of nodes—are solvable within minutes by SCIP-Jack. The hard set comprises PUC instances (Rosseti et al., 2001), most of which cannot be solved within one hour by SCIP-Jack and even lack known optima. BKS are determined by taking the best value between SCIP-Jack's one-hour primal bound and published solutions from SteinLib or Vienna-GEO (Leitner et al., 2014). We also highlight the 2018 PACE Challenge[20] as a useful benchmark with varied difficulty levels.

---

[13]http://www.lac.inpe.br/~lorena/instancias.html
[14]http://stegger.net/somala/index.html
[15]https://github.com/phil85/GB21-MH
[16]https://github.com/Lei-Kun/FJSP-benchmarks
[17]https://github.com/INFORMSJoC/2021.0326
[18]https://steinlib.zib.de/steinlib.php
[19]https://dimacs11.zib.de/organization.html
[20]https://github.com/PACE-challenge/SteinerTree-PACE-2018-instances

Training data includes two generation strategies. The first generator corresponds to the hardest instances in PUC (Rosseti et al., 2001), which constructs graphs from hypercubes with randomly sampled (perturbed) edge weights. We generate 100 training instances for neural solvers and 10 validation instances for LLMs across dimensions 6–10. The second, based on GeoSteiner (Juhl et al., 2018), samples 25,000-node graphs from a unit square. We include 15 such instances (10 for neural solvers, 5 for LLMs)[21], and add 45 adapted TSPLib instances (Juhl et al., 2018) to the neural training set. The LLM training set also serves as the validation set for neural solvers.

## C   IMPLEMENTATION DETAILS

### C.1   NEURAL SOLVERS

**DiffUCO, SDDS.**   The DiffUCO/SDDS checkpoints used in our evaluation are taken directly from the official repository[22] and correspond to the models trained on the RB-Large dataset. For MIS (easy and hard), we increase the number of inference steps to 50, while for MDS-easy we revert to the default of 3 steps. Both models encounter out-of-memory issues on the MDS-hard set.

**RLNN.**   We use the official checkpoint trained on RB-[800–1200][23] for all evaluations. All inference hyperparameters follow the original paper, except that we increase the number of inference steps to 100,000 on both MIS-easy and MIS-hard to fully utilize the 1-hour time budget.

**DIFUSCO.**   We use the official checkpoint trained on TSP-10000[24]. Decoding uses the greedy + 2-OPT heuristic, and all other inference parameters follow the configuration used in the original paper on TSP-10000.

**LEHD.**   We use the optimal TSP and CVRP checkpoints from the official repository[25]. Decoding employs the Parallel Local Reconstruction (PRC) heuristic. Instead of fixing the number of PRC iterations, we continue iterating until the 3600-second time budget is exhausted.

**DeepACO.**   For CVRP, we evaluate the official CVRP500 checkpoint[26]. After the neural construction phase, we follow the standard protocol and continue decoding using HGS until the time limit is reached.

**SIL.**   We use the default checkpoints from the official repository for TSPLib (trained on TSP-1000) and CVRPLib (trained on CVRP-1000) [27]. Similar to LEHD, decoding uses PRC and is iterated until the solving time budget is exhausted.

**tMDP, SORREL.**   We use the official checkpoints for the CFLP task from the tMDP[28] and SORREL[29] repositories. For tMDP, we follow the DFS-based variant.

**GCNN.**   For CFLP, GCNN is trained on 100,000 strong-branching samples collected from 10,000 instances. For CPMP, it is trained on 50,000 samples collected from 1,000 instances. Both training procedures follow the methodology of Gasse et al. (2019).

**IL-LNS, CL-LNS.**   Training data is constructed from local-branching trajectories on 200 instances. We follow the default protocol, using 20% of variables to define the large neighborhood, and keep all remaining hyperparameters identical to those in the official implementation[30].

---

[21] http://www.geosteiner.com/instances/
[22] https://github.com/ml-jku/DIffUCO
[23] https://github.com/Shengyu-Feng/RLD4CO
[24] https://github.com/Edward-Sun/DIFUSCO
[25] https://github.com/CIAM-Group/NCO_code/tree/main/single_objective/LEHD
[26] https://github.com/henry-yeh/DeepACO
[27] https://github.com/CIAM-Group/SIL
[28] https://github.com/lascavana/rl2branch
[29] https://github.com/Shengyu-Feng/SORREL
[30] https://github.com/facebookresearch/CL-LNS

**MPGN, L-RHO.**   Since the FJSP instances used in our experiments are already compatible with those evaluated by MPGN[31] and L-RHO[32], we adopt their exact hyperparameter settings, including the 450 training instances generated by Li et al. (2025a).

## C.2   LLM SOLVERS

**Self-Refine**   In our implementation, we run 64 iterations. In each iteration, the LLM receives the previous best-performing code and its dev-set evaluation results, then generates the next code. We use o4-mini with a medium reasoning budget and default sampling parameters. The dev evaluation timeout is 300s, although the LLM is prompted to write algorithms for a 3600s timeout. After 64 iterations, we evaluate the best dev-set code on the test set with a 3600s timeout.

**FunSearch**   We follow the official FunSearch implementation and modify the prompt to fit our tasks. We set the number of islands to 10, functions per prompt to 2, the reset period to 2 hours, and run 64 iterations with a 300s dev evaluation timeout. After 64 iterations, we evaluate the best dev-set code on the test set with a 3600s timeout.

**ReEvo**   We follow the official ReEvo implementation and modify the prompt to fit our tasks. We set the population size to 10, initial population size to 4, mutation rate to 0.5, and run 64 iterations with a 300s dev evaluation timeout. After 64 iterations, we evaluate the best dev-set code on the test set with a 3600s timeout.

## C.3   EXAMPLE PROMPT

Our query prompts basically consist of two parts: the description of the problem background and the starter code for LLM to fill in. The following is an example prompt on TSP.

---

**The evaluation example**

**Problem Description**

The Traveling Salesman Problem (TSP) is a classic combinatorial optimization problem where, given a set of cities with known pairwise distances, the objective is to find the shortest possible tour that visits each city exactly once and returns to the starting city. More formally, given a complete graph $G = (V, E)$ with vertices V representing cities and edges E with weights representing distances, we seek to find a Hamiltonian cycle (a closed path visiting each vertex exactly once) of minimum total weight.

**Starter Code**

```python
def solve(**kwargs):
    """
    Solve a TSP instance.

    Args:
        - nodes (list): List of (x, y) coordinates representing
            cities in the TSP problem
                    Format: [(x1, y1), (x2, y2), ..., (xn, yn)]

    Returns:
        dict: Solution information with:
            - 'tour' (list): List of node indices representing the
                solution path
                            Format: [0, 3, 1, ...] where numbers
                                are indices into the nodes list
    """
    # Your function must yield multiple solutions over time, not
        just return one solution
```

---

[31]https://github.com/wrqccc/FJSP-DRL
[32]https://github.com/mit-wu-lab/l-rho

```
    # Use Python's yield keyword repeatedly to produce a stream of
        solutions
    # Each yielded solution should be better than the previous one
    while True:
        yield {
            'tour': [],
        }
```

## D    DETAILED RESULTS

Table 4–11 present the detailed results for the evaluated methods in Section 4. A result is marked with * if the method suffers from the out-of-memory or timeout issue before obtaining a feasible solution (assigned a primal gap 1 and runtime 3600 seconds) on any instance in this benchmark. Note that the geometric mean and standard deviation is reported for the solving time.

Table 4: Comparative Results on MIS.

| MIS | Easy | | Hard | |
|---|---|---|---|---|
| Method | Gap ↓ | Time ↓ | Gap ↓ | Time ↓ |
| KaMIS | $1.51 \pm 0.43\,\%$ | $223 \pm_\times 5.62\,\text{s}$ | $\mathbf{2.65} \pm 0.81\,\%$ | $274 \pm_\times 1.29\text{s}$ |
| DiffUCO | $9.54 \pm 9.82\,\%$ | $154 \pm_\times 1.18\,\text{s}$ | $6.45 \pm 1.43\,\%$ | $19 \pm_\times 1.29\,\text{s}$ |
| SDDS | $11.85 \pm 12.47\,\%$ | $223 \pm_\times 1.14\,\text{s}$ | $5.24 \pm 1.12\,\%$ | $27 \pm_\times 1.22\,\text{s}$ |
| RLNN | $6.29 \pm 8.81\,\%$ | $532 \pm_\times 3.24\,\text{s}$ | $6.31 \pm 2.76\,\%$ | $1064 \pm_\times 1.65\,\text{s}$ |
| FunSearch | $1.87 \pm 3.63\,\%$ | $3600 \pm_\times 1.00\,\text{s}$ | $4.97 \pm 0.92\,\%$ | $3600 \pm_\times 1.00\,\text{s}$ |
| Self-Refine | $\mathbf{1.30} \pm 3.89\,\%$ | $3600 \pm_\times 1.00\,\text{s}$ | $4.02 \pm 0.97\,\%$ | $3600 \pm_\times 1.00\,\text{s}$ |
| ReEvo | $1.44 \pm 4.05\,\%$ | $3600 \pm_\times 1.00\,\text{s}$ | $4.81 \pm 0.95\,\%$ | $3600 \pm_\times 1.00\,\text{s}$ |

Table 5: Comparative Results on MDS.

| MDS | Easy | | Hard | |
|---|---|---|---|---|
| Method | Gap ↓ | Time ↓ | Gap ↓ | Time ↓ |
| Gurobi | $\mathbf{0.00} \pm 0.00\,\%$ | $3600 \pm_\times 1.00\,\text{s}$ | $\mathbf{0.63\%} \pm 2.74\,\%$ | $3600 \pm_\times 1.00\,\text{s}$ |
| DiffUCO | $71.86 \pm 21.56\,\%$ | $54 \pm_\times 26.86\,\text{s}$ | $^*100.00 \pm 0.00\,\%$ | $^*3600 \pm_\times 1.00\,\text{s}$ |
| SDDS | $66.21 \pm 12.80\,\%$ | $54 \pm_\times 27.01\,\text{s}$ | $^*100.00 \pm 0.00\,\%$ | $^*3600 \pm_\times 1.00\,\text{s}$ |
| FunSearch | $41.83 \pm 48.67\,\%$ | $3600 \pm_\times 1.00\,\text{s}$ | $95.21 \pm 11.43\,\%$ | $3600 \pm_\times 1.00\,\text{s}$ |
| Self-Refine | $6.19 \pm 4.42\,\%$ | $3600 \pm_\times 1.00\,\text{s}$ | $5.71 \pm 3.49\,\%$ | $3600 \pm_\times 1.00\,\text{s}$ |
| ReEvo | $7.52 \pm 4.50\,\%$ | $3600 \pm_\times 1.00\,\text{s}$ | $5.81 \pm 5.24\,\%$ | $3600 \pm_\times 1.00\,\text{s}$ |

Table 6: Comparative Results on TSP.

| TSP | Easy | | Hard | |
|---|---|---|---|---|
| Method | Gap ↓ | Time ↓ | Gap ↓ | Time ↓ |
| LKH-3 | **0.03** ± 0.05 % | 65 ±$_\times$ 8.90 s | **2.89** ± 1.58 % | 21 ±$_\times$ 6.69 s |
| LEHD | 10.23 ± 9.37 % | 487 ±$_\times$ 4.20 s | *76.84 ± 34.23 % | *1347 ±$_\times$ 1.63 s |
| DIFUSCO | 4.19 ± 1.20 % | 555 ±$_\times$ 2.45 s | *69.04 ± 45.57 % | *2850 ±$_\times$ 1.66 s |
| SIL | 2.51 ± 1.56 % | 3600 ±$_\times$ 1.00 s | 21.34 ± 34.23 % | 3600 ±$_\times$ 1.00 s |
| FunSearch | 6.79 ± 5.80 % | 3600 ±$_\times$ 1.00 s | 35.82 ± 25.62 % | 3600 ±$_\times$ 1.00 s |
| Self-Refine | 6.29 ± 5.35 % | 3600 ±$_\times$ 1.00 s | 32.00 ± 17.44 % | 3600 ±$_\times$ 1.00 s |
| ReEvo | 5.65 ± 6.16 % | 3600 ±$_\times$ 1.00 s | 37.77 ± 38.57 % | 3600 ±$_\times$ 1.00 s |

Table 7: Comparative Results on CVRP.

| CVRP | Easy | | Hard | |
|---|---|---|---|---|
| Method | Gap ↓ | Time ↓ | Gap ↓ | Time ↓ |
| HGS | **0.11** ± 0.18 % | 3600 ±$_\times$ 1.00 s | 6.74 ± 2.50 % | *3600 ±$_\times$ 1.00 s |
| LEHD | 1.97 ± 0.92 % | 893 ±$_\times$ 1.74 s | *100.00 ± 0.00% | *3600 ±$_\times$ 1.00 s |
| DeepACO | 4.42 ± 1.56 % | 50 ±$_\times$ 1.64 s | *27.69 ± 36.18 % | *3333 ±$_\times$ 1.12s |
| SIL | 10.90 ± 8.17 % | 3600 ±$_\times$ 1.00 s | 11.35 ± 4.46 % | 3600 ±$_\times$ 1.00 s |
| FunSearch | 5.27 ± 3.70 % | 3600 ±$_\times$ 1.00 s | **6.52** ± 2.67 % | 3600 ±$_\times$ 1.00 s |
| Self-Refine | 3.86 ± 1.63 % | 3600 ±$_\times$ 1.00 s | 27.50 ± 6.19 % | 3600 ±$_\times$ 1.00 s |
| ReEvo | 7.16 ± 3.42 % | 3600 ±$_\times$ 1.00 s | 10.01 ± 2.83 % | 3600 ±$_\times$ 1.00 s |

Table 8: Comparative Results on CFLP.

| CFLP | Easy | | Hard | |
|---|---|---|---|---|
| Method | Gap ↓ | Time ↓ | Gap ↓ | Time ↓ |
| Gurobi | **0.00** ± 0.00 % | 308 ±$_\times$ 1.93 s | **0.01** ± 0.02 % | 3136 ±$_\times$ 1.34 s |
| tMDP | 3.54 ± 3.14 % | 3581 ±$_\times$ 1.00 s | 55.35 ± 21.7 % | 3600 ±$_\times$ 1.00 s |
| SORREL | 3.46 ± 2.51% | 3600 ±$_\times$ 1.00 s | 55.35 ± 21.7 % | 3600 ±$_\times$ 1.00 s |
| GCNN | 3.22 ± 3.10 % | 3551 ±$_\times$ 1.07 s | 55.35 ± 21.7 % | 3600 ±$_\times$ 1.00 s |
| FunSearch | 7.31 ± 0.75 % | 3600 ±$_\times$ 1.00 s | 7.41 ± 3.26 % | 3600 ±$_\times$ 1.00 s |
| Self-Refine | 27.08 ± 10.79 % | 3600 ±$_\times$ 1.00 s | 24.93 ± 21.56 % | 3600 ±$_\times$ 1.00 s |
| ReEvo | 12.89 ± 1.70 % | 3600 ±$_\times$ 1.00 s | 12.79 ± 6.40 % | 3600 ±$_\times$ 1.00 s |

Table 9: Comparative Results on CPMP.

| CPMP | Easy | | Hard | |
|---|---|---|---|---|
| Method | Gap ↓ | Time ↓ | Gap ↓ | Time ↓ |
| GB21-MH | **0.53** ± 0.49 % | 541 ±$_\times$ 8.49 s | **0.32** ± 0.37 % | 3600 ±$_\times$ 1.00 s |
| IL-LNS | *80.57 ± 36.75 % | *2636 ±$_\times$ 1.92 s | *100.00 ± 0.00 % | *3600 ±$_\times$ 1.00 s |
| CL-LNS | *81.45 ± 36.21 % | *2649 ±$_\times$ 1.92 s | *100.00 ± 0.00 % | *3600 ±$_\times$ 1.00 s |
| GCNN | *42.91 ± 28.66 % | *2143 ±$_\times$ 3.68 s | *100.00 ± 0.00 % | *3600 ±$_\times$ 1.00 s |
| FunSearch | 3.96 ± 3.77 % | 3600 ±$_\times$ 1.00 s | *77.32 ± 41.06 % | *3600 ±$_\times$ 1.00 s |
| Self-Refine | 2.84 ± 2.57 % | 3600 ±$_\times$ 1.00 s | *74.05 ± 39.50 % | *3600 ±$_\times$ 1.00 s |
| ReEvo | 3.40 ± 3.14 % | 3600 ±$_\times$ 1.00 s | *70.64 ± 43.61 % | *3600 ±$_\times$ 1.00 s |

Table 10: Comparative Results on FJSP.

| FJSP | Easy | | Hard | |
|---|---|---|---|---|
| Method | Gap ↓ | Time ↓ | Gap ↓ | Time ↓ |
| CPLEX | **0.00** ± 0.00 % | 702 ±$_\times$ 17.01 s | **0.01** ± 0.04 % | 3600 ±$_\times$ 1.00 s |
| MPGN | 12.78 ± 4.04 % | 9 ±$_\times$ 4.26 s | 1.50 ± 0.85 % | 69 ±$_\times$ 1.90 s |
| L-RHO | 27.20 ± 12.97 % | 21 ±$_\times$ 1.87 s | 1.03 ± 0.86 % | 58 ±$_\times$ 2.49 s |
| FunSearch | 5.05 ± 3.57 % | 3600 ±$_\times$ 1.00 s | 12.10 ± 2.90 % | 3600 ±$_\times$ 1.00 s |
| Self-Refine | 6.66 ± 2.48 % | 3600 ±$_\times$ 1.00 s | 1.14 ± 1.27 % | 3600 ±$_\times$ 1.00 s |
| ReEvo | 5.61 ± 2.78 % | 3600 ±$_\times$ 1.00 s | 2.16 ± 1.72 % | 3600 ±$_\times$ 1.00 s |

Table 11: Comparative Results on STP.

| STP | Easy | | Hard | |
|---|---|---|---|---|
| Method | Gap ↓ | Time ↓ | Gap ↓ | Time ↓ |
| SCIP-Jack | **0.00** ± 0.00 % | 22 ±$_\times$ 5.43 s | **0.50** ± 0.62 % | 717 ±$_\times$ 26.70 s |
| RL | 14.00 ± 3.31 % | 31 ±$_\times$ 8.40 s | 13.10 ± 6.52 % | 1 ±$_\times$ 4.44 s |
| SL | 14.00 ± 3.31 % | 31 ±$_\times$ 8.40 s | 13.10 ± 6.52 % | 1 ±$_\times$ 4.44 s |
| FunSearch | 8.29 ± 5.44 % | 3600 ±$_\times$ 1.00 s | 5.82 ± 4.86 % | 3600 ±$_\times$ 1.00 s |
| Self-Refine | 11.23 ± 6.04 % | 3600 ±$_\times$ 1.00 s | 6.93 ± 3.96 % | 3600 ±$_\times$ 1.00 s |
| ReEvo | 14.36 ± 3.53 % | 3600 ±$_\times$ 1.00 s | 10.03 ± 6.43 % | 3600 ±$_\times$ 1.00 s |

# E  EFFICIENCY ANALYSIS OF NEURAL SOLVERS

Neural solvers are typically motivated as fast heuristics that avoid the heavy computation of exact classical solvers. **However, existing evaluations often overlook that many classical solvers can also operate as fast heuristics when given restricted time budgets.** To illustrate this point, we take TSP as an example and compare commonly used fast-mode configurations of both classical and neural methods. Specifically, we include LKH-3 under the POPMUSIC setting (Taillard & Helsgaun, 2019) with 1,000 trials, DIFUSCO with 50 inference steps and greedy decoding, LEHD with greedy decoding, and SIL with 10 Parallel Local Reconstruction (PRC) steps. The comparative results on several TSPLib instances across different scales are shown in Table 12.

Table 12: Comparison between the fast version of classical and neural solvers on TSPLib instances across different scales.

| TSP-easy Instances | LKH-3 | | LEHD | | DIFUSCO | | SIL | |
|---|---|---|---|---|---|---|---|---|
| | Gap ↓ | Time ↓ | Gap ↓ | Time ↓ | Gap ↓ | Time ↓ | Gap ↓ | Time ↓ |
| pr1002 | **0.70**% | 0.2s | 17.02% | 6s | 4.25% | 3s | 0.82% | 27s |
| fl1400 | **0.40**% | 0.3s | 10.52% | 6s | 28.37% | 8s | 5.53% | 31s |
| pr2392 | **0.68**% | 0.4s | 10.97% | 18s | 14.80% | 15s | 3.46% | 38s |
| pcb3038 | **0.68**% | 1s | 14.70% | 20s | 11.85% | 31s | 3.27% | 28s |
| fnl4461 | **0.76**% | 1s | 16.00% | 81s | 3.50% | 34s | 2.64% | 39s |
| rl5915 | **1.43**% | 2s | 9.42% | 169s | 3.31% | 52s | 5.41% | 39s |
| rl11894 | **2.25**% | 4s | 27.56% | 1204 | 31.47% | 130s | 6.41% | 37s |
| usa13509 | **1.53**% | 4s | 41.75% | 1765s | 33.14% | 164s | 11.33% | 42s |
| brd14051 | **1.39**% | 4s | 29.19% | 1981s | 30.37% | 167s | 6.14% | 39s |
| d15112 | **0.12**% | 5s | 26.49% | 2464s | 3.20% | 278s | 5.21% | 40s |
| d18512 | **1.22**% | 5s | 29.64% | 4767s | 30.67% | 266s | 5.56% | 41s |

The results indicate that neural solvers remain substantially less effective than LKH-3 when all methods are run in their fast configurations. As the instance size grows from 1,002 to 18,512 nodes, LKH-3 requires only five additional seconds while still delivering near-optimal solutions. In contrast,

DIFUSCO needs an extra six minutes, and LEHD requires over an hour, merely to obtain a single feasible tour. SIL is notably more scalable than both LEHD and DIFUSCO—its runtime increases much more modestly, reflecting meaningful progress in ML-based solvers. However, its optimality gap remains large and it does not show a clear advantage over LKH-3 in runtime or solution quality.

## F  ADDITIONAL ANALYSES ON NON-EUCLIDEAN CHALLENGES IN TSP

To further highlight the difficulty posed by non-Euclidean CO instances—a challenge largely overlooked in current ML4CO evaluations—we incorporate an additional Asymmetric TSP (ATSP) benchmark composed of non-Euclidean (non-metric) instances. These instances originate from real-world datasets (Jordan Srour & van de Velde, 2013) that span stacker-crane operations, transportation and routing tasks, robotic motion planning, and data-compression problems. Dataset statistics and evaluation results are reported in Table 13 and Table 14, respectively. RRNCO (Son et al., 2025), a recent neural solver designed for real-world ATSP instances, is used for evaluation.

Table 13: Summary of ATSP instances.

| Problem | Test Set Sources | Attributes | Easy Set | Hard Set |
|---------|------------------|------------|----------|----------|
| ATSP | Jordan Srour & van de Velde (2013) | Instances | 31 | 33 |
| | | Cities | 131 | 323–932 |

Table 14: Comparative Results on ATSP.

| ATSP | Easy | | Hard | |
|------|------|------|------|------|
| Method | Gap $\downarrow$ | Time $\downarrow$ | Gap $\downarrow$ | Time $\downarrow$ |
| LKH-3 | $\mathbf{0.00} \pm 0.00\,\%$ | $1927 \pm_\times 1.31\,\mathrm{s}$ | $\mathbf{0.08} \pm 0.09\,\%$ | $705 \pm_\times 2.52\,\mathrm{s}$ |
| RRNCO | $1.42 \pm 0.69\,\%$ | $3600 \pm_\times 1.00\,\mathrm{s}$ | $15.46 \pm 7.88\,\%$ | $3600 \pm_\times 1.00\,\mathrm{s}$ |
| FunSearch | $0.00 \pm 0.00\,\%$ | $3600 \pm_\times 1.00\,\mathrm{s}$ | $3.52 \pm 5.26\,\%$ | $3600 \pm_\times 1.00\,\mathrm{s}$ |
| Self-Refine | $0.50 \pm 0.81\,\%$ | $3600 \pm_\times 1.00\,\mathrm{s}$ | $10.94 \pm 11.16\,\%$ | $3600 \pm_\times 1.00\,\mathrm{s}$ |

It can be seen that although the largest ATSP instance, which is from the hard set, only contains 932 nodes, it is way challenging than the easy (metric) TSP instances at the small scale ($\sim 1000$ nodes), where the former could not be solved to optimum within 1 hour while the latter could be solved to optimum in seconds.

## G  DISCUSSIONS IN SOLVER DEPLOYMENT

Beyond solution quality and solving time, we observe substantial variation in deployment cost across solver families. In general, neural solvers rely heavily on GPUs and require significantly more memory than classical solvers, making them more expensive to deploy in practice. LLM-based solvers, while consuming considerable API credits or training resources during the evolutionary or development phase, incur relatively low deployment cost during inference once the solver is finalized.

