# OpenReview forum: "FrontierCO: Real-World and Large-Scale Evaluation of Machine Learning Solvers for Combinatorial Optimization"
_ICLR.cc/2026/Conference — ICLR 2026 Poster_

### Official Review · Reviewer_zcmM · 2025-10-23

**Soundness:** 3
**Presentation:** 3
**Contribution:** 3
**Rating:** 6
**Confidence:** 3

**Summary:**

This paper proposed a systematic, large scale benchmark for machine learning for combinatorial optimization field. The benchmark collected a wide range of datasets, and standardized training pipelines, and exhibited insightful findings of current ML4CO methods compared to standard solvers.

**Strengths:**

A large scale and wide benchmark makes a lot of sense for ML4CO field. The selection of problems are reasonable. The best known solution is a good guidance for CO practitionerrs. And the findings about current ML vs solvers are important guidelines for future research.

**Weaknesses:**

First of all, there is no clear weakness of the paper. There are some small concerns that I have:
- The problem instances in table 1 are all collected from previous literature or competitions, which might limit the originality of the benchmark.
- The standardized training and validation makes sense to some extent, but it may not be fair comparison. e.g., some methods may subsume scaling law and performs better with more training data, while other methods might be suitable for data scarcity but stagnant with more training data. Also, baselines such as GNNs seem incomparable to LLMs, as the modality is totally different. So it is not convincing to me if there exists a standardized training pipeline.
- I think the deployment difficulty of various methods should also be considered, such as GPU hours, for real world applications.

**Questions:**

- In equation 2, why is the primal gap set to 1 when $cost \cdot c* < 0$?
- There also exists other CO benchmarks, e.g., https://github.com/Thinklab-SJTU/ML4CO-Bench-101, so what is your core competitive strength as another CO benchmark paper?

---

> ### Author Response · Authors · 2025-11-21
>
> > There also exists other CO benchmarks, e.g., ML4CO-Bench-101. What is your core competitive strength as another CO benchmark paper?
>
> ML4CO-Bench-101 is a valuable recent effort and we have added it to Section 6 of the revised manuscript. Conceptually, ML4CO-Bench-101 aggregates existing ML4CO benchmarks, which mostly contain medium-scale synthetic instances. FrontierCO is designed to complement this by introducing large-scale real-world datasets that are widely used in the classical CO community but have not been incorporated into ML4CO pipelines. FrontierCO also provides LLM-ready natural-language problem descriptions and unified evaluation protocols across neural, LLM, and classical solvers. **These elements—real-world scale, broader coverage, and LLM compatibility—form the core competitive strengths of our benchmark.**
>
> | Dataset            | Portion of New Datasets | Portion of Real-World Datasets | Largest Instance (# Variables) | Supports LLM Evaluation? |
> |--------------------|--------------------------|----------------------------------|--------------------------------|---------------------------|
> | ML4CO-Bench-101    | 0%                       | 9%                               | 10k                            | No                        |
> | FrontierCO         | 75%                      | 69%                              | 10M                            | Yes                       |
>
>
> > The problem instances in Table 1 are all collected from previous literature or competitions, which might limit the originality of the benchmark.
>
> While FrontierCO draws evaluation instances from established CO libraries (TSPLib, DIMACS, CFLP testbeds, etc.), it is not a simple aggregation. A major contribution of FrontierCO is that it **provides curated, labeled training and validation sets for these real-world instances**, enabling ML methods to be applied to data that previously lacked the supervision needed for training neural solvers or refining LLM agents (see Section 2.5 and Appendix B). Existing datasets in classical CO benchmarks typically offer only test instances without standardized training resources, making ML4CO evaluation inconsistent or impractical. By pairing real-world evaluation tasks with **newly generated, standardized, and size-appropriate training/validation data**, FrontierCO bridges this long-standing gap between classical CO datasets and ML-based solver development.
>
> > The standardized training and validation makes sense to some extent, but it may not be a fair comparison since some models benefit from scaling laws while others saturate early. Also, GNNs seem incomparable to LLMs because the modalities are different.
>
> We address fairness from two perspectives.
> - (1) **Training**: FrontierCO does not enforce a single fixed training set size, since different model families have different computational limits. Instead, we use training setups consistent with each method’s original implementation, ensuring that the comparison reflects each solver’s intended operating regime rather than engineering differences.
> - (2) **Validation**: Once trained, all solvers—GNNs, LLMs, and classical heuristics—are evaluated under a unified inference protocol with identical instance formats, runtime limits, and resource constraints. From the evaluation standpoint, this makes heterogeneous models directly comparable.
>
> > The deployment difficulty of various methods (e.g., GPU hours) should also be considered for real-world applications.
>
> We agree this is an important dimension. In Appendix I, we now include a discussion of deployment considerations such as approximate GPU hours and setup overheads. While our main goal is to provide a unified performance evaluation, these additional notes highlight the computational trade-offs relevant for practitioners.
>
> > In Equation (2), why is the primal gap set to 1 when cost(x;s)⋅c*<0?
>
> This case arises when the returned solution has an objective value with the opposite sign of the reference optimum, meaning the solution is worse than a trivial one. In such situations, the cost cannot be meaningfully normalized against c* (since worse the solution, smaller the primal gap), and the output is treated as invalid. Following standard practice in CO evaluation, we assign a primal gap of 1 for any invalid or sign-inconsistent output for rigorous definition (though it does not happen to be present in our evaluated problems). We clarified this behavior in Appendix F  with concrete examples.

---

> > ### Comment · Reviewer_zcmM · 2025-11-25
> >
> > Thanks for your effort for the reply!
> > Now my questions are answered and the concerns are all addressed. I believe this work has some novelty and significance compared with existing benchmark papers. Given some proper engineering and wrapping work to the paper, the CO and ML4CO community will benefit a lot in the era of large scale training and LLMs.
> > Therefore I would like to raise my score a bit.

---

> > > ### Author Response · Authors · 2025-11-25
> > >
> > > Thank you for the encouraging feedback and for increasing your score!

---

### Official Review · Reviewer_VRL2 · 2025-10-29

**Soundness:** 3
**Presentation:** 3
**Contribution:** 2
**Rating:** 4
**Confidence:** 4

**Summary:**

This paper presents a standardized benchmark for the evaluation of machine learning–based solvers for eight combinatorial optimization problems. It provides a comparison of 16 ML-based solvers against state-of-the-art classical methods. The study offers some insights, revealing both the current limitations and the future potential of ML-based solvers in this domain.

**Strengths:**

1. The methodology of the paper is good and well presented.

2. Developing standardized benchmarks for evaluating ML-based CO solvers is necessary for progress in this field.

3. Proposed benchmark cover routing, graph, location, set, and scheduling CO problems.  However, another scheduling problem would be welcome.

**Weaknesses:**

I believe that some of the claims in this paper are too strong and not supported by the current state of research in this domain.

It is true, in general, that many neural solvers rely on attention mechanisms and suffer from well-known attention bottlenecks, and this paper points out that addressing these limitations is an important direction for future research. However, many works have already explored ways to mitigate these bottlenecks, which are entirely overlooked in this work. Some relevant works are listed below.

1. The work by Luo et al. [1] specifically addresses the attention bottleneck problem in routing tasks. It directly challenges the claim in Section 5.1 that neural solvers are still far from being comparable to state-of-the-art classical solvers. In fact, their approach outperforms both LKH3 and HGS on large-scale CVRP instances.

2. The scalability analysis in this work considers only a few (computationally expensive) models and concludes that NCO solvers have problems with scalability. An example in this work is LEHD, and it, as its name implies, relies on a Heavy Decoder architecture, which is very costly. However, there are significantly faster alternatives, for example, the method proposed in [2], and their experiments show that it is over 200× faster than LKH3, with only a minor drop in performance.

3. The claim that most neural solvers are based on GNNs is not entirely accurate. Current SOTA neural solvers primarily employ attention mechanisms rather than GNNs - including the majority of models you evaluate in this work.

4. The discussion and claims about capturing global structure using Euclidean and non-Euclidean problems are based on a single problem -  STP. I am not sure that is enough to conclude the ability of neural solvers to capture the global graph structure.

5. I fully support the claim that most existing evaluations are limited to 2D Euclidean instances. However, unfortunately, this work did not take any steps to address this limitation - the benchmarks still exclude non-Euclidean instances for TSP and CVRP, despite the growing number of recent works focusing on this aspect. Including such instances would significantly strengthen the arguments and make the evaluation more comprehensive, especially given the stated goal of building a “real-world evaluation of NCO.” In practice, all real-world routing problems involve non-Euclidean distances. I therefore suggest adding the instances from [2], or any other real-world datasets, to your benchmark set.

[1] Luo et al. Boosting Neural Combinatorial Optimization for Large-Scale Vehicle Routing Problems, ICLR 2024

[2] Son et al. Neural Combinatorial Optimization for Real-World Routing, arXiv:2503.16159

**Questions:**

In addition to my remarks in the weaknesses section, I have the following questions:

1. I did not fully understand your procedure for generating training instances. Let’s take the descriptions of the TSP and VRP training datasets as examples. You first mention that instances are uniformly sampled from a unit square. However, you later state that DIMACS instances with 1,000 nodes are used for the LLM dataset. Does this mean you created two separate datasets - one for “classic” NCO solvers and another for LLM-based methods?

2. Additionally, you mention that 15 validation VRP instances are used “for LLMs,” but there is no information about how many training instances were generated, nor any details about the validation dataset for TSP. Why do TSP instances have 1,000 nodes, while CVRP instances go only up to 500 nodes? Do you consider 15 validation instances sufficient for a reliable evaluation?

I would appreciate further clarification on these points, as well as on the training datasets for the other problems.

Additionally, I do not fully understand your claim that you provide regenerated training and validation datasets to eliminate inconsistencies found in previous works, and the limitation that existing models are trained on synthetic instances. Your TSP and CVRP datasets appear to be identical to those used in prior works - synthetic instances with node coordinates sampled from the unit square, demands drawn from [1, 9], and capacity set to 50.

3. You mention that an asterisk (*) indicates that the method encountered out-of-memory or timeout issues. In that case, how was the optimality gap computed for those models if they did not produce any solution?

---

> ### Public Comment · ~Fu_Luo1 · 2025-11-20
>
> Dear Authors of Paper 13404:
>
> Thank you very much for presenting such an exciting benchmark. However, I would like to point out that there may be some possible concerns:
>
> 1. Issues with the division of some data sets. For example, you involve Golden instances (Golden et al., 1998) in the easy set of CVRP. However, as I know, the Golden instances 1-8 have route-length restrictions, i.e., the distance traveled by each vehicle is limited to a specific number. However, this paper does not clearly state this.
>
> 2. Additionally, I agree with Reviewer VRL2's concern on this setting: "For any infeasible solution, we assign a primal gap of 1". If an instance cannot be solved due to some specific issues, such as out-of-memory, it should not be included in the calculation of the average gap.
>
> Best Regards

---

> > ### Author Response · Authors · 2025-11-21
> >
> > Thank you for your comment!
> >
> > 1. This is a good point, we have now added the clarification in Appendix B.
> >
> > 2. We beg to disagree on this one. The key purpose of our work is to **understand the potential of different solvers in really advancing the human intelligence (that is why we include so many unsolved open instances and have LLMs in)**. Therefore, scalability would be an important factor. If we always exclude these instances, the metric would not reflect our research purpose.

---

> ### Author Response · Authors · 2025-11-21
>
> > However, another scheduling problem would be welcome.
> > I therefore suggest adding the instances from [2], or any other real-world datasets, to your benchmark set.
>
> Thanks for the suggestion. **We have added a new set of real-world Asymmetric TSP (ATSP) instances** from [3], including 33 easy and 9 hard cases across stacker crane, routing, robot motion and data compression applications. Appendix H now reports solver performance on these datasets, along with a comparison to the TSP instances, addressing this limitation directly.
>
> [3] F. Jordan Srour, Steef van de Velde, Are Stacker Crane Problems easy? A statistical study, Computers & Operations Research, In Press, Corrected Proof, Available online 1 July 2011, ISSN 0305-0548, DOI: 10.1016/j.cor.2011.06.017.
>
> > I believe that some of the claims in this paper are too strong and not supported by the current state of research in this domain … directly challenges the claim in Section 5.1 that neural solvers are still far from being comparable to state-of-the-art classical solvers [1][2]
>
> **We want to clarify that this apparent contradiction arises from the evaluation protocol**, not from empirical disagreement.
>
> - First, real-world instances (as in our FrontierCO) are **more diverse in terms of the size and distribution**. Although we have observed the advantage of [1] against classical solvers on some instances (e.g., Leuven2 in the hard set) in FrontierCO, this advantage is not consistent. Our newly added results in Table 6,7,15 (part of them shown below) still demonstrate the inferior performance of [1,2] against SOTA classical solvers (3600s limit).
>
> - Second, works [1,2] report batched inference times for neural solvers (e.g., evaluating 128 testing instances in parallel and dividing the total time by 128). **This measures GPU throughput, not the expected solving time of a single instance**. In our **single-instance, single-thread evaluation**, both [1,2] are slower  than classical solvers.
>
> - Finally, the classical solvers' ability to serve as a fast heuristic beyond exact solvers are ignored by existing evaluation (see Section E).
>
> ### Evaluation results with 1-hour time budget
>
> | CVRP   | Gap on the Easy Set | Gap on the Hard Set |
> |-----------|----------------------|----------------------|
> | HGS       | **0.11%**               |  **6.74%**             |
> | SIL [1]   | 10.90%              | 11.35%              |
>
>
> | ATSP    | Gap on the Easy Set | Gap on the Hard Set |
> |------------|----------------------|----------------------|
> | LKH-3      | **0.00%**                  | **0.08%**               |
> | RRNCO [2]  | 1.42%              | 15.46%              |
>
> [1] Luo et al. Boosting Neural Combinatorial Optimization for Large-Scale Vehicle Routing Problems, ICLR 2024
>
> [2] Son et al. Neural Combinatorial Optimization for Real-World Routing, arXiv:2503.16159
>
> > It is true, in general, that many neural solvers rely on attention mechanisms and suffer from well-known attention bottlenecks
> > The work by Luo et al. [1] specifically addresses the attention bottleneck problem in routing tasks
>
> Thank you a lot for drawing our attention to this strong baseline, and we have included it as the new evaluation model on TSP and CVRP. We do observe a better performance in general against neural solvers we evaluated, but still, as shown in the results above, less effective than classical solvers on real-world benchmarks.
>
> > The scalability analysis considers only computationally expensive models (e.g., LEHD). However, there are significantly faster alternatives such as [2], which is over 200× faster than LKH3 with minor performance drops.
>
> Following your suggestion, we now include RRNCO [2] in our ATSP evaluation in Table 15. However, as we mentioned above, its evaluation is problematic and actually slower than LKH3. Since it uses a dense graph as the input, it also encounters OOM issues on instances with >5000 nodes.
>
> > The claim that most neural solvers are based on GNNs is not entirely accurate. Current SOTA solvers employ attention mechanisms, including many you evaluate.
>
> To avoid confusion, we clarified the terminology used in the broader graph-learning literature. In that literature, **attention-based graph architectures are conventionally categorized as GNNs** because they operate on graph-structured inputs and follow the standard graph-processing paradigm. Representative examples include:
> - Graph Attention Networks (GAT; ICLR 2018)
> - Graphormer (NeurIPS 2021)
> - SAN (Structure-Aware Transformer; NeurIPS 2020)
> - GPS (Graph Positioning System; NeurIPS 2022)
> - HGT (Heterogeneous Graph Transformer; WWW 2020)
>
> These architectures are widely regarded as GNN variants because they operate on graph-structured inputs, even though their aggregation functions are attention-based. We updated the discussion in the paper to make this conventional usage explicit.

---

> ### Author Response · Authors · 2025-11-21
>
> > The discussion about global structure relies only on STP, which may not be sufficient for general conclusions.
>
> We extended the analysis to non-Euclidean TSP (ATSP) instances in Table 16. Neural solvers exhibit significantly larger degradation on these datasets, consistent with the behavior observed in STP. Including two structurally distinct families reinforces the conclusion that many neural CO solvers struggle when locality assumptions break down.
>
> >I do not understand the instance-generation procedure (uniform sampling vs. DIMACS). Does this imply separate datasets for different solvers?
>
> To clarify, FrontierCO **does not enforce a single universal training-set size across all solvers**. Different neural and LLM-based methods have fundamentally different training-time scalability and memory limits, as reflected in their original papers. Enforcing a single large training set would unfairly disadvantage some methods and distort the comparison. **Instead, FrontierCO enforces the same instances distribution for training (or validation) instances.**
>
> All training data for TSP are drawn from the same underlying distribution (e.g., uniform sampling in the unit square). The number and size of instances differ by solver, reflecting the computational limits reported in their original implementations. For simplicity, we use a subset of DIMACS instances that also follows the uniform point distribution as our LLM validation instances.  We revised Section 2.2 to state this more explicitly.
>
> > Why do TSP instances have 1,000 nodes, while CVRP instances go only up to 500 nodes?
>
> CVRP is known to be more challenging than TSP, here we reduce the size to control its difficulty when used in LLM validation. But we still use CVRP1000 for LEHD, following its choice.
>
> > Why does CVRP use only 15 validation instances? Is this sufficient?
>
> This choice is consistent with common practice in both neural CO (e.g., DIFUSCO uses 8) and classical VRP studies (typically 10–20). This size offers diverse coverage while remaining computationally manageable.
>
> > I do not understand the claim about “regenerated” datasets.
>
> As before, we revised the text in Section 2.2 to clarify that each method is trained and validated on instances drawn from the same distribution, as to enable the fair comparison between neural and LLM solvers (they used to be trained/developed on instances from different distribution in previous works).
>
> > How is the gap computed for OOM or timeout?
>
> Following Section 3.1, if a solver fails to produce a feasible solution (OOM, timeout, or invalid output), we assign primal gap = 1 and time = 3600s, following standard CO convention. This is applied consistently across all tasks.

---

> > ### Comment · Reviewer_VRL2 · 2025-11-24
> >
> > Thank you very much for your response.
> >
> > Adding ATSP instances significantly strengthens these benchmarks.
> >
> > However, I still do not understand the motivation for using a primal gap of 1 for non-existing solutions, and I cannot agree with it. I have never seen this before, and it seems very contradictory to me. I read the added Appendix F and saw that you refer to a PhD thesis from 2006, which is not a particularly strong reference.
> >
> > Another point of confusion is that in your manuscript, you usually express the primal gap as a percentage, yet for non-existing solutions, you define it as 1. Is that meant to represent 100%?
> >
> > To illustrate why this seems wrong: imagine we have one solver that can successfully solve all instances with an optimality gap of 1.01 (i.e., 101%), and another solver that produces trash - in other words, it cannot find any feasible solution. According to your benchmark, the second solver would be considered better. This makes no sense.
> >
> > If I have understood everything correctly, your benchmark results appear to be incorrect and potentially misleading, and I cannot support the acceptance of this work.

---

> > > ### Author Response · Authors · 2025-11-24
> > > **Clarification in the Misunderstanding of our Primal Gap Definition**
> > >
> > > Thank you for the feedback! We are glad to see the ATSP instances address your concern.
> > >
> > > Regarding the primal-gap concern, the **misunderstanding stems from mixing our normalized primal gap with the optimality gap used in routing literature** such as [1].
> > > - Our metric (applicable to general CO) is  $$\frac{|\mathrm{obj}-\mathrm{optim}|}{\max\{(|\mathrm{obj}|,|\mathrm{optim}|)\}}$$
> > > - whereas the optimality gap (routing centric) in [1] is
> > >  $$\frac{\mathrm{obj}-\mathrm{optim}}{\mathrm{optim}}$$
> > >
> > > By construction, our primal gap lies strictly in $[0,1]$ for any feasible solution.  Thus the reviewer’s hypothetical “101\% gap” cannot occur: feasible solutions always yield $pg\leq 1$, and infeasible ones always receive $pg=1$.
> > >
> > > **Regarding the literature supporting the normalized primal-gap metric, the formulation**
> > >
> > > $$\frac{|\mathrm{obj}-\mathrm{optim}|}{\max\{(|\mathrm{obj}|,|\mathrm{optim}|)\}}$$
> > >
> > > **is first proposed but not specific to Berthold’s 2006 thesis** [2]. It is the standard metric used in classical Mixed-Integer Programming research and solvers, including Berthold et al.’s analyses of primal heuristics [3], Achterberg et al.’s rounding and propagation heuristics [4], and the 11th DIMACS Challenge rules [5]. The same metric is used in modern neural-enhanced MIP systems—e.g., Nair et al. [6], Chmiela et al. [7], Huang et al. [8]. We have updated Appendix F to reflect these references and clarify that this metric follows longstanding conventions in both classical and learning-based combinatorial optimization.
> > >
> > > [1] Luo et al. Boosting Neural Combinatorial Optimization for Large-Scale Vehicle Routing Problems, ICLR 2024
> > >
> > > [2] Berthold et al.. Primal heuristics for mixed integer programs. PhD thesis, Zuse Institute Berlin (ZIB), 2006.
> > >
> > > [3] Berthold et al. Measuring the impact of primal heuristics. Operations Research Letters 41(6):611–614.
> > >
> > > [4] Achterberg et al. Rounding and propagation heuristics for mixed integer programming. In Operations Research Proceedings 2011.
> > >
> > > [5] 11th DIMACS challenge. https://dimacs11.zib.de/docs/CompetitionRules-141119.pdf
> > >
> > > [6] Nair et al. Solving Mixed Integer Programs Using Neural Networks. 2020.
> > >
> > > [7] Chmiela et al. Learning to Schedule Heuristics in Branch-and-Bound. NeurIPS 2021.
> > >
> > > [8] Huang et al. Searching large neighborhoods for integer linear programs with contrastive learning. ICML 2023.

---

> > > > ### Comment · Reviewer_VRL2 · 2025-11-25
> > > >
> > > > Good, thank you for the clarification!
> > > >
> > > > Now your methodology makes sense, although I still prefer to explicitly see the quantity of unfeasible solutions.
> > > >
> > > > Thank you for the clarifications; I will increase my score.

---

> > > > > ### Author Response · Authors · 2025-11-25
> > > > >
> > > > > Thank you for the thoughtful reconsideration and for raising your score! Your feedback is greatly appreciated, we would add the quantity of unfeasible solutions in the revised version.

---

### Official Review · Reviewer_uR1C · 2025-11-01

**Soundness:** 3
**Presentation:** 3
**Contribution:** 2
**Rating:** 6
**Confidence:** 5

**Summary:**

The paper introduces FRONTIERCO, a benchmark for evaluating ML-based combinatorial optimization (CO) solvers under real-world structure and extreme scale. It targets three long-standing limitations in prior NCO evaluations: small “toy” instances, synthetic data that miss structural diversity, and inconsistent baselines/splits. The benchmark spans eight CO problems across routing, facility location, scheduling, graphs, and Steiner trees, drawing test instances from competitions and public repositories (e.g., TSPLib, DIMACS). For training/validation, the authors provide standardized synthetic generators for neural solvers and separate dev sets for LLM agents to avoid leakage; test evaluation is strictly on real-world instances. A unified protocol enforces a one-hour evaluation cap and a scale-invariant primal-gap metric with an explicit infeasibility policy.

**Strengths:**

[Problem motivation & clarity]
The introduction tightly argues why synthetic, small-scale evaluations have over-estimated ML performance and motivates a real-world, frontier-scale benchmark. The easy/hard split, data provenance, and scale claims are clearly articulated, making the problem and goals unambiguous.

[Benchmark design & practicality]
The benchmark aggregates eight diverse CO tasks with real-world test instances and provides standardized BKS and synthetic training/dev resources. The one-hour cap and hardware normalization (single CPU core for all, single A6000 for neural solvers) produce a pragmatic, comparable setting. The scale-invariant primal-gap definition (including the infeasibility policy) is a strong choice for cross-task fairness.

[Breadth and diagnostic depth of evaluation]
Sixteen ML solvers spanning neural, hybrid, and LLM agents are compared to top classical solvers. Beyond averages, the paper probes why gaps occur: distribution shift from synthetic training to structurally diverse test sets; neural OOM/latency; and LLM variability/safety issues. The STP Euclidean vs. non-Euclidean experiment is a crisp test of “global structure” capture.

[Insightful, constructive negative results]
The study shows that neural modules can improve weak baselines (e.g., DIFUSCO vs. 2-OPT; GCNN vs. SCIP) yet still trail strong classical SOTA, and it identifies promising pockets where LLM agents surpass classical SOTA (e.g., MIS-easy, CVRP-hard) while cautioning about instability. These balanced findings are valuable for the community.

**Weaknesses:**

1. The paper’s treatment of metric edge cases lacks technical clarity. While the primal-gap policy is defined—including handling of negative or zero costs and infeasible outputs—there are few concrete examples that cover both minimization and maximization settings. This matters because tasks can differ in objective sign and scale, and without worked examples readers may interpret the same numeric gap differently across problems. The gap definition section would benefit from illustrative cases; as is, reproduction and extension to new tasks or cost conventions can be confusing.

2. Reporting granularity is limited, creating questions about numerical consistency. The main text emphasizes aggregate gaps, but offers few per-instance distributions, confidence intervals, or paired significance tests. This makes it difficult to assess the reliability of observed improvements or claimed LLM “wins.” In the results tables and figures, richer dispersion statistics and paired comparisons on shared instances would substantiate robustness and provide clearer selection guidance for practitioners.

3. Transparency around compute and training budgets is insufficient. Although the evaluation setup standardizes a one-hour cap and hardware, the paper provides only limited per-method training details for neural baselines (e.g., epochs, steps, seeds, hyperparameters) and sparse accounting of sampling budgets for LLM agents. This obscures fairness and cost-effectiveness comparisons across approaches. The evaluation and appendices should enumerate these budgets; without them, reproducibility and cost–benefit assessment are hindered.

4. The scope and novelty of LLM-based algorithms are not framed sharply enough. Word-clouds and qualitative analysis suggest that the agents largely recompose known metaheuristics such as simulated annealing and large-neighborhood search. What is missing is an explicit separation between retrieval/recombination and genuinely novel algorithmic invention. In the Discussion (§5.3), clarifying this boundary would calibrate expectations for future “agentic” progress and reduce the risk of overstating algorithmic novelty.

5. Finally, statistical robustness and safety considerations are underdeveloped. The LLM agents exhibit high variance across runs—sometimes generating very different strategies—and can trigger OOM or other resource issues during iterative “evolving.” The paper does not provide a systematic variance/seed study or a resource-safety protocol (e.g., per-sample CPU/GPU caps). These gaps, noted across the results and discussion, leave practical adoption uncertain without clearer guardrails on reliability and operational cost.

**Questions:**

[Clarify and exemplify the metric]
Provide a boxed, self-contained definition of the primal-gap for both minimization and maximization tasks, with fully worked examples that cover negative and zero objective values as well as an infeasible output (e.g., explicitly show a case yielding gap = 1 with time = 3600s). Accompany this with a concise table mapping each task to its objective definition and sign convention so readers cannot misinterpret gap magnitudes across problems with differing scales or signs.

[Make evaluation and training budgets fully transparent]
Enumerate complete training and sampling budgets so cost–performance trade-offs are clear and reproducible. For neural methods, report dataset sources and sizes, epochs/steps, batch sizes, learning-rate schedules, regularizers, number of seeds, total GPU hours, and peak memory. For LLM agents, disclose prompts/templates, sampling parameters (temperature, top-p), number of samples/iterations, wall-clock per sample, and aggregate CPU/GPU hours; include fair wall-clock comparisons to classical solvers under the same evaluation limits.

[Strengthen statistical reporting and robustness]
Report mean ± standard deviation (or confidence intervals) per test set and run paired statistical tests (e.g., Wilcoxon) on shared instances to substantiate improvements. Release raw per-instance outcomes and add scatter plots (gap versus time), clearly marking infeasible or failed runs. Include sensitivity analyses: for neural models, vary model size and any reduction techniques; for LLM agents, vary the number of samples/iterations and compare tool-use versus no-tool settings.

[Probe integration with strong classical components]
Move beyond “neural versus weak heuristic” and evaluate “neural + strong heuristic,” for example by plugging neural move-selection into LKH-3 or HGS variants. Measure whether such hybrids consistently lift near-SOTA classical solvers across tasks and scales, thereby clarifying the realistic contribution of learned components.

[Expand structure-shift diagnostics]
Generalize the Euclidean/non-Euclidean analysis beyond STP to graph problems such as MIS/MDS (e.g., SAT-induced versus random graphs) and to routing (metric versus non-metric variants). Report performance deltas, learning-curve contrasts, and any systematic failure modes to better characterize how models cope with structural distribution shifts.

[Improve reproducibility aids]
Release concise pseudocode for each evaluation harness (neural and LLM), including the hidden evaluation API signature to eliminate ambiguity in adapter interfaces. Publish all seeds, configuration files, and scripts required to regenerate figures and tables from raw outputs. Add a short “gotchas” section documenting environment pitfalls (e.g., Python/BLAS/MKL versions) that can materially affect results.

---

> ### Author Response · Authors · 2025-11-21
>
> > While the primal-gap policy is defined—including handling of negative or zero costs and infeasible outputs—there are few concrete examples that cover both minimization and maximization settings.
>
> To address this, we revised Section 2.1 to explicitly state that all maximization tasks are converted into minimization form by negating their objectives, ensuring a unified primal-gap definition. Appendix F now includes worked examples covering optimal, negative, and infeasible outputs for both minimization and maximization tasks. Thanks for the comment.
>
>
> > Reporting granularity is limited, creating questions about numerical consistency.
>
> Thank you for your suggestion. Following your suggestion, we now include standard deviations alongside means in Appendix D. We have also provided more instance-wise analyses in Table 12.
>
> > Enumerate complete training and sampling budgets so cost–performance trade-offs are clear and reproducible.
>
> Appendix C now provides all training and sampling details. For the problems where we used official checkpoints, we also detail the one we used. These additions ensure that all reported results are fully reproducible.
>
> > Move beyond “neural versus weak heuristic” and evaluate “neural + strong heuristic,”
> > For neural models, vary model size and any reduction techniques; for LLM agents, vary the number of samples/iterations and compare tool-use versus no-tool settings.
>
> Our benchmark is designed to evaluate methods under a consistent protocol rather than re-optimize each solver. To avoid introducing solver-specific engineering, we preserve the canonical implementations from the original papers and adjust only when necessary for compatibility within our unified evaluation setup.
>
> >  Word-clouds and qualitative analysis suggest that the agents largely recompose known metaheuristics such as simulated annealing and large-neighborhood search. What is missing is an explicit separation between retrieval/recombination and genuinely novel algorithmic invention.
>
> In our analysis, retrieval/recombination refers to the algorithms that can be directly mapped to known heuristics  (e.g., SA-style acceptance rules, LNS destroy–repair patterns, simple neighborhood moves). In contrast, algorithmic invention would require qualitatively new operators or search dynamics that cannot be reduced to existing templates. We have further clarified this on Section 5.3.
>
>
> > Release concise pseudocode for each evaluation harness (neural and LLM), including the hidden evaluation API signature… Publish all seeds, configuration files, and scripts required to regenerate figures and tables from raw outputs.
>
> We have added the missing details. Appendix C provides the full LLM evaluation template, and the supplementary package contains API signature, all seeds, configuration files, and scripts used to reproduce all tables and figures.
>
>
> > Generalize the Euclidean/non-Euclidean analysis beyond STP
>
> Thank you for the suggestion, we have now added one more category of data for non-Euclidean/non-metric TSP instances in Appendix H.

---

### Author Response · Authors · 2025-11-21
**Overall response**

We thank all reviewers for the thoughtful and constructive feedback. We are glad that the reviewers recognize our key contributions, including (i) introducing diverse, large-scale, and real-world CO instances to the ML4CO community, (ii) standardizing comparisons across neural, LLM-based, and classical solvers, and (iii) offering diagnostic findings that highlight both progress and limitations of current ML-based solvers.

Below we summarize how we addressed the major shared concerns:
- **Inclusion of non-Euclidean TSP instances (uR1C, VRL2).**
We added a new category of non-Euclidean TSP (ATSP) instances (33 easy and 9 hard real-world cases) covering stacker-crane, routing tasks, robotic motion, and data compression applications. Appendix H now includes full evaluations and comparisons to Euclidean-based benchmarks.
- **Clarification of the primal-gap definition (uR1C, zcmM).**
Section 2.1 now explicitly states the conversion of maximization tasks into minimization form. Appendix F provides a unified boxed definition with worked examples covering optimal, negative-sign, and infeasible outputs.
- **Expanded training, validation, and evaluation details (uR1C, VRL2).**
Appendix A,C now report data generation procedures, training budgets, hyperparameters, sampling and iteration budgets, and resource constraints for all neural and LLM-based methods.
- **Improved explanation of training-set construction (VRL2).**
We clarified that FrontierCO does not enforce a single universal training-set size because different solvers have fundamentally different training scalability limits. Each solver uses dataset sizes consistent with its original implementation, while all data are drawn from a shared underlying distribution. This preserves fairness across diverse solver families.
- **Additional related work and contextualization (VRL2, zcmM).**
We incorporated the suggested related works—including Luo et al. Son et al. and ML4CO-Bench-101—and clarified how FrontierCO complements existing benchmarks by providing curated, labeled training and validation sets for real-world CO tasks.

These revisions substantially improve the clarity, completeness, and reproducibility of the benchmark.

---

### Comment · Area_Chair_pQGd · 2025-11-24
**Discussion Period**

Dear reviewers,

The discussion period is now open. Please use the “Official Comments” to engage in discussions about each other's reviews and the authors' rebuttal, and update your assessments or comments as appropriate.

Did the authors' rebuttal adequately address your concerns? We kindly ask that you update your reviews based on these discussions and your evaluation of the rebuttal, even if your overall assessment remains unchanged.

Thank you all for your contributions.

Best regards, AC

---

### Meta-Review · Area_Chair_B2c2 · 2025-12-29

**Summary:**

This work proposes a new benchmark called FRONTIERCO for evaluating ML-based combinatorial optimization solvers under realistic, large-scale, and structurally diverse problem settings. It demonstrates that while neural solvers can improve weak heuristics and LLMs can sometimes discover novel algorithm combinations, they still fall short of state-of-the-art classical solvers, especially on hard real-world instances.

The reviewers originally had mixed scores (6,6,4) and raised concerns regarding the overly strong claims without sufficient supporting evidence, inadequate scalability analysis, missing discussion/comparison with related work, unclear experimental settings, and the need for further clarification. After the rebuttal, two reviewers indicated that their concerns had been adequately addressed and thus raised their scores to 8 and 6 respectively, resulting in a final set scores (8,6,6) for this work.

I read this paper in detail and agree with the reviewers that this work is a timely and valuable contribution to the ML4CO community, and therefore recommend accepting this work.

**Reviewer Concerns:**

I believe most concerns have been properly addressed by the rebuttal. A few remaining minor concerns could be:


1. Quantity of Unfeasible Solutions: I agree with Reviewer VRL2 and the Public Commenter that it would be better to explicitly report the quantity of infeasible solutions. Setting the primal gap to 1 for an infeasible solution does not feel intuitive to me, even though it has been used in prior work. From my understanding, an infeasible solution is much worse than the worst feasible solution. The authors have committed to including the quantity of infeasible solutions in the revised paper.

2. Neural Solver as GNN: On one hand, I understand the authors' rationale for categorizing attention-based models as GNNs, and I think referencing a relevant work [1] (a current technical version of an influential 2020 blog post) would help support this claim. On the other hand, I am not fully convinced of the necessity of this categorization for the ML4CO community, as many works on neural solvers never consider the GNN perspective when employing attention-based models.

[1] Chaitanya K. Joshi, Transformers are Graph Neural Networks, arXiv:2506.22084.

**Reviewer Scores:**

According to the author-reviewer discussion, if the reviewers had been able to participate fully in the discussion,  I believe Reviewer VRL2 would have increased the score from 4 to 6 and Reviewer zcmM from 6 to 8. Consequently, the final score for this work could reasonably be expected to be (8, 6, 6).

---

### Decision · Program_Chairs · 2026-01-26

Accept (Poster)